# Smoothed Energy Guidance: Guiding Diffusion Models with Reduced Energy Curvature of Attention

**Susung Hong**[*]
University of Washington

## Abstract

Conditional diffusion models have shown remarkable success in visual content generation, producing high-quality samples across various domains, largely due to classifier-free guidance (CFG). Recent attempts to extend guidance to unconditional models have relied on heuristic techniques, resulting in suboptimal generation quality and unintended effects. In this work, we propose Smoothed Energy Guidance (SEG), a novel training- and condition-free approach that leverages the energy-based perspective of the self-attention mechanism to enhance image generation. By defining the energy of self-attention, we introduce a method to reduce the curvature of the energy landscape of attention and use the output as the unconditional prediction. Practically, we control the curvature of the energy landscape by adjusting the Gaussian kernel parameter while keeping the guidance scale parameter fixed. Additionally, we present a query blurring method that is equivalent to blurring the entire attention weights without incurring quadratic complexity in the number of tokens. In our experiments, SEG achieves a Pareto improvement in both quality and the reduction of side effects. The code is available at https://github.com/SusungHong/SEG-SDXL.

## 1 Introduction

Diffusion models [12, 45, 46] have emerged as a promising tool for visual content generation, producing high-quality and diverse samples across various domains, including image [38, 40, 42, 8, 13, 30, 2, 24, 9, 29, 34, 33, 4, 41, 5, 20, 22], video [11, 50, 23, 18, 15, 3, 19, 44], and 3D generation [36, 27, 6, 26, 49, 43, 48, 16]. The success of these models can be largely attributed to the use of classifier-free guidance (CFG) [14], which enables sampling from a sharper distribution, resulting in improved sample quality. However, CFG is not applicable to unconditional image generation, where no specific conditions are provided, creating a disparity between the capabilities of text-conditioned sampling and sampling without text. This disparity results in a restriction in application, *e.g.*, synthesizing images with ControlNet[51] without a text prompt (see the last two columns of Fig. 1).

Recent literature [17, 1] has attempted to decouple CFG and image quality by extending guidance to general diffusion models, leveraging their inherent representations [25, 32, 17]. Self-attention guidance (SAG) [17] proposes leveraging the intermediate self-attention map of diffusion models to blur the input pixels and provide guidance, while perturbed attention guidance (PAG) [1] perturbs the attention map itself by replacing it with an identity attention map. Despite these efforts, these methods rely on heuristics to make perturbed predictions, resulting in unintended effects such as smoothed-out details, saturation, color shifts, and significant changes in the image structure when given a large guidance scale. Notably, the mathematical underpinnings of these unconditional guidance approaches are not well elucidated.

---

[*]This work was mostly done at Korea University.

38th Conference on Neural Information Processing Systems (NeurIPS 2024).

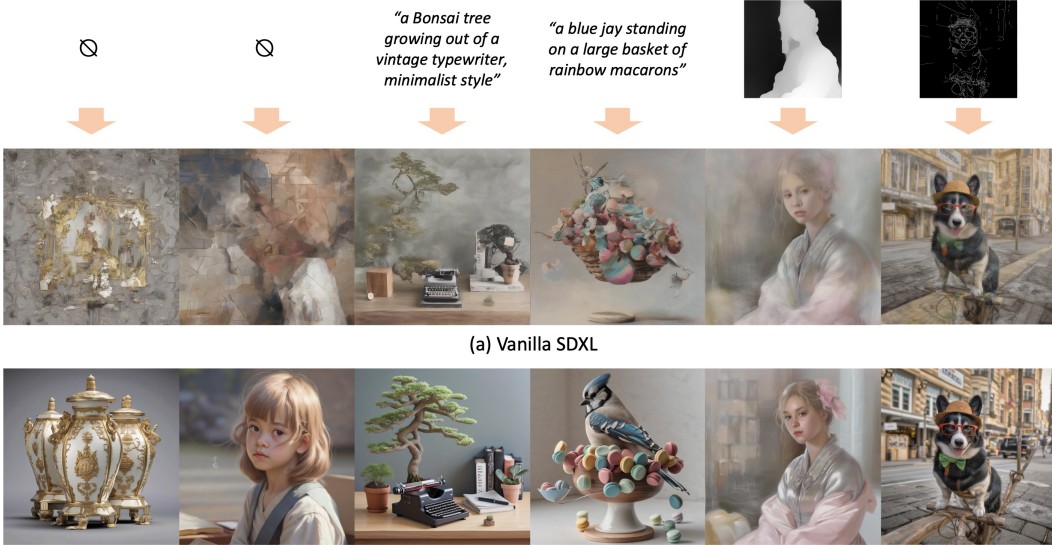

"a Bonsai tree growing out of a vintage typewriter, minimalist style"

"a blue jay standing on a large basket of rainbow macarons"

(a) Vanilla SDXL

(b) SDXL + Smoothed Energy Guidance (Ours)

Figure 1: Teaser. (a) Images sampled from vanilla SDXL [35] without any guidance. (b) Images sampled with Smoothed Energy Guidance (Ours). ∅ denotes that there is no condition given. With various input conditions, and even without any, SEG supports the diffusion model in generating plausible and high-quality images without any training.

In this work, we approach the objective from an energy-based perspective of the self-attention mechanism, which has been previously explored based on its close connection to the Hopfield energy [39, 31, 7]. Specifically, we start from the definition of the energy of self-attention, where performing a self-attention operation is equivalent to taking a gradient step. In light of this, we propose a tuning- and condition-free method that **reduces the curvature** of the underlying energy function by directly blurring the attention weights, and then leverages the output as the negative prediction. We call this method **Smoothed Energy Guidance (SEG)**.

SEG does not merely rely on the guidance scale parameter that cause side effects when its value becomes large. Instead, we can continuously control the original and maximally attenuated curvature of the energy landscape behind the self-attention by simply adjusting the parameter of the Gaussian kernel, with the guidance scale parameter fixed. Additionally, we introduce a novel query blurring technique, which is equivalent to blurring the entire attention weights without incurring quadratic cost in the number of tokens.

We validate the effectiveness of SEG throughout the various experiments without and with text conditions, and ControlNet [51] trained on canny and depth maps. Based on the attention modulation, SEG results in less structural change from the original prediction compared to previous approaches [17, 1], while achieving better sample quality.

## 2 Preliminaries

### 2.1 Diffusion models

Diffusion models [12, 45, 46] are a class of generative models that generate data through an iterative denoising process. The process of adding noise to an image $\mathbf{x}$ over time $t \in [0, T]$ is governed by the forward stochastic differential equation (SDE):

$$d\mathbf{x} = \mathbf{f}(\mathbf{x}, t)dt + g(t)d\mathbf{w}, \tag{1}$$

where $\mathbf{f}$ and $g$ are predefined functions that determine the manner in which the noise is added, and $d\mathbf{w}$ denotes a standard Wiener process.

Correspondingly, the denoising process can be described by the reverse SDE:

$$d\mathbf{x} = [\mathbf{f}(\mathbf{x}, t) - g(t)^2 \nabla_{\mathbf{x}} \log p_t(\mathbf{x})]dt + g(t)d\bar{\mathbf{w}}, \tag{2}$$

where $\nabla_{\mathbf{x}} \log p_t(\mathbf{x})$ represents the score of the noisy data distribution and $d\bar{\mathbf{w}}$ denotes the standard Wiener process for the reversed time.

Diffusion models are trained to approximate the score function with $\mathbf{s}_\theta(\mathbf{x}, t) \approx \nabla_{\mathbf{x}} \log p_t(\mathbf{x})$. To generate an image based on a condition $c$, *e.g.*, a class label or text, one simply needs to train diffusion models to approximate the conditional score function with $\mathbf{s}_\theta(\mathbf{x}, t, c) \approx \nabla_{\mathbf{x}} \log p_t(\mathbf{x}|c)$ and replace $\nabla_{\mathbf{x}} \log p_t(\mathbf{x})$ with it in the denoising process. To enhance the quality and faithfulness of the generated samples, classifier-free guidance (CFG) [14] is widely adopted. Accordingly, the reverse process becomes:

$$dx = [\mathbf{f}(\mathbf{x}, t) - g(t)^2(\gamma_{\text{cfg}}\mathbf{s}_\theta(\mathbf{x}, t, c) - (\gamma_{\text{cfg}} - 1)\mathbf{s}_\theta(\mathbf{x}, t))]dt + g(t)d\bar{\mathbf{w}}. \tag{3}$$

Here, $\mathbf{s}_\theta(\mathbf{x}, t)$ is learned by dropping the label by a certain proportion, and $\gamma_{\text{cfg}}$ is a hyperparameter that controls the strength of the guidance. Intuitively, CFG helps us to sample from sharper distribution by conditioning on a class label or text.

## 2.2 Energy-based view of attention mechanism

The attention mechanism [47], which has been widely adopted in diffusion models [12], has been interpreted through the lens of energy-based models (EBMs) [31, 39, 7], especially through its close connection with the Hopfield energy [7, 39]. In the modern (continuous) Hopfield network, the attention operation can be derived based on the concave-convex procedure (CCCP) from the following energy function [39]:

$$E(\boldsymbol{\xi}) = -\text{lse}(\mathbf{X}\boldsymbol{\xi}^\top) + \frac{1}{2}\boldsymbol{\xi}\boldsymbol{\xi}^\top, \tag{4}$$

where $\boldsymbol{\xi} \in \mathbb{R}^{1 \times d}$, $\mathbf{X} \in \mathbb{R}^{N \times d}$, and lse stands for the *log-sum-exp function*, defined as $\text{lse}(\mathbf{v}) := \log\left(\sum_{i=1}^{N} e^{v_i}\right)$. The quadratic term acts as a regularizer to prevent $\boldsymbol{\xi}$ from exploding [39], while $-\text{lse}(\mathbf{X}\boldsymbol{\xi}^\top)$ penalizes misalignment between $\mathbf{X}$ and $\boldsymbol{\xi}$.

Mathematically, it turns out that the attention mechanism is equivalent to the update rule of the modern Hopfield network [7, 39]. Specifically, inspired by the Hopfield energy in (4), and noticing that the first term depends on the attention weights, we propose the following energy function for entire self-attention weights in diffusion models:

**Definition 2.1** (Energy Function for Self-Attention). *Let* $\mathbf{Q} \in \mathbb{R}^{(HW) \times d}$ *be a matrix of query vectors and* $\mathbf{K} \in \mathbb{R}^{(HW) \times d}$ *be a matrix of key vectors, where* $H$, $W$, *and* $d$ *represent the height, width, and dimension, respectively. Let* $\mathbf{A} \in \mathbb{R}^{(HW) \times (HW)} := \mathbf{Q}\mathbf{K}^\top$. *The energy function with respect to entire self-attention weights in diffusion models is defined as:*

$$E(\mathbf{A}) := \sum_{i=1}^{H}\sum_{j=1}^{W} E'(\mathbf{a}_{:(i,j)}), \quad E'(\mathbf{a}) := -\text{lse}(\mathbf{a}) = -\log\left(\sum_{k=1}^{H}\sum_{l=1}^{W} e^{a_{(k,l)}}\right). \tag{5}$$

Note that to explicitly denote the spatial dimension, we use the subscript $(x, y)$ to represent the index of a row or column of the matrices. Despite using the definition in (5) for the rest of the paper for simplicity, we additionally discuss the dual case, where we use the swapped indexing, in Appendix B.

This view leads us to an important intuition: the attention operation can be seen as a minimization step on the energy landscape, considering that the first derivative represents the softmax operation which also appears in the attention operation. Building upon this intuition, we argue that Gaussian blurring on the attention weights modulates the underlying landscape to have less curvature, and we demonstrate this in the following sections by analyzing the second derivatives.

## 3 Method

Our aim is to theoretically derive the effect of Gaussian blur applied on the attention weights, which in the end attenuates the curvature of the underlying energy function. Then, utilizing this fact, we develop attention-based drop-in diffusion guidance that enhances the quality of the generated samples, regardless of whether an explicit condition is given. In Section 3.1, we claim some useful properties of Gaussian blur: that it preserves mean, reduces variance, and thus decreases the lse value. In

Section 3.2, we find that the curvature of the energy landscape is attenuated by the attention blur operation, leading naturally to a blunter prediction for guidance. And finally, in Section 3.3, built upon this fact, we define Smoothed Energy Guidance (SEG) and propose the equivalent query blurring method, which can perform attention blurring while avoiding quadratic complexity in the number of tokens.

## 3.1 Gaussian blur to attention weights

In this section, we derive some important properties of the Gaussian blur with the aim of figuring out the variation of the energy landscape. To this end, we start from some mathematical underpinnings on applying Gaussian blur to attention weights.

A 2D Gaussian filter is a convolution kernel that uses a 2D Gaussian function to assign weights to neighboring pixels. The 2D Gaussian function is defined as:

$$G(x, y) = \frac{1}{2\pi\sigma^2} e^{-\frac{(x-\mu_x)^2 + (y-\mu_y)^2}{2\sigma^2}}$$

where $\mu_x$ and $\mu_y$ are the means in the $x$ and $y$ directions, and $\sigma$ is the standard deviation. The 2D Gaussian filter possesses symmetry, *i.e.*, $G(x, y) = G(-x, -y)$, and normalization, *i.e.*, $\iint G(x, y)dxdy = 1$. In practice, we use a discretized version of the Gaussian filter with a finite kernel size depending on $\sigma$, normalized to sum to 1.

**Lemma 3.1.** *Spatially applying a 2D Gaussian blur to the attention weights* $\mathbf{a} := \mathbf{Q}\mathbf{k}^\top$ *preserves the average* $\mathbb{E}_{i,j}[a_{(i,j)}]$. *In addition, the variance monotonically decreases every time we apply the Gaussian blur.*

*Proof sketch.* Applying a 2D Gaussian filter to the attention weights $a_{(i,j)}$ yields the blurred values $\tilde{a}_{(i,j)}$:

$$\tilde{a}_{(i,j)} = \sum_{m=-k}^{k} \sum_{n=-k}^{k} G(m, n) \cdot a_{(i+m, j+n)}$$

where $k$ is the filter size, $G(m, n)$ is the Gaussian filter value at position $(m, n)$, and $a_{(i+m, j+n)}$ is the attention weight at position $(i+m, j+n)$. Since the Gaussian filter is symmetric and normalized, it can be shown that the mean of the blurred attention weights is equal to the mean of the original attention weights. Similarly, we can show that the variance monotonically decreases when we apply a 2D Gaussian filter. See Appendix A.1 for the complete proof.

Note that this fact also implies that blurring with a Gaussian filter with a larger standard deviation causes a greater decrease in the variance of attention weights. This is because a Gaussian filter with a larger standard deviation can always be represented as a convolution of two filters with smaller standard deviations, due to the associativity of the convolution operation.

Finally, we show that applying a 2D Gaussian blur to attention weights increases the lse value in (5), *i.e.*, increases the energy in (5). This provides a bit of intuition about the underlying energy landscape, yet it is more prominently utilized in the claims in the following sections.

**Lemma 3.2.** *Applying a 2D Gaussian blur to attention weights* $\mathbf{a} := \mathbf{Q}\mathbf{k}^\top$ *increases the* lse *term when we consider the second-order Taylor series approximation of the exponential function around the mean* $\mu := \mathbb{E}_{i,j}[a_{(i,j)}]$. *Consequently, the maximum is achieved when the attention is uniform, i.e.,* $a_{(i,j)} = a_{(k,l)} \ \forall i, j, k, l$. *This corresponds to the case when we apply the Gaussian blur with* $\sigma \to \infty$.

*Proof sketch.* Applying the second-order Taylor series approximation around the mean $\mu$, and using Proposition 3.1, we show that the second-order approximation of $\text{lse}(\mathbf{a})$ is larger than or equal to that of $\text{lse}(\tilde{\mathbf{a}})$. Subsequently, we introduce Lagrange multipliers to find the maximum, which gives us the result, $a_{(i,j)} = a_{(k,l)} \ \forall i, j, k, l$. We leave the full proof in Appendix A.2.

## 3.2 Analysis of the energy landscape

In this section, we demonstrate that applying a 2D Gaussian blur to the attention weights before the softmax operation results in computing the updated value with reduced curvature of the underlying energy function. To this end, we analyze the Gaussian curvature before and after blurring the attention weights. This is closely related to the Hessian of the energy function.

**Theorem 3.1.** *Let the attention weights be defined as* $\mathbf{a} := \mathbf{Q}\mathbf{k}^\top$. *Consider the energy function in* (5). *Then, applying a Gaussian blur to the attention weights* $\mathbf{a}$ *before the softmax operation results in the attenuation of the Gaussian curvature of the underlying energy function where gradient descent is performed.*

*Proof sketch.* Let $\mathbf{H}$ denote the Hessian of the original energy function, *i.e.*, the derivative of the negative softmax, and $\tilde{\mathbf{H}}$ denote the Hessian of the new energy function associated with blurred attention weights. Furthermore, let $b_{ij}$ denote the $i$-th row, $j$-th column entry in the Toeplitz matrix $\mathbf{B}$ representing the Gaussian blur. Calculating the derivatives, we have the elements of the Hessians, $h_{ij} = (\xi(\mathbf{a})_i - \delta_{ij})\xi(\mathbf{a})_j$ and $\tilde{h}_{ij} = (\xi(\tilde{\mathbf{a}})_i - \delta_{ij})\xi(\tilde{\mathbf{a}})_j b_{ij}$. Using Lemmas 3.1 and 3.2 and under reasonable assumptions, we observe that $|\det(\mathbf{H})| > |\det(\tilde{\mathbf{H}})|$, which implies that the minimization step is performed on a smoother energy landscape with attenuated Gaussian curvature. The full proof is in Appendix A.3.

To provide more intuition about what is actually happening and how we utilize this property in the later section, it is intriguing to consider the attenuating effect on the curvature in analogy to classifier-free guidance (CFG). CFG uses the difference between the prediction based on the sharper conditional distribution and the prediction based on the smoother unconditional distribution to guide the sampling process. By analogy, we propose a method to make the landscape of the energy function smoother to guide the sampling process, as opposed to the original (sharper) energy landscape.

From a probabilistic perspective, the energy is associated with the likelihood of the attention weights in terms of the Boltzmann distribution conditioned on a given configuration, *i.e.*, the feature map. Blurring the attention weights diminishes this likelihood as shown in Lemma 3.2, and also reduces the curvature of the distribution as shown in Theorem 3.1.

### 3.3 Smoothed energy guidance for diffusion models

Based on the above observation that the Gaussian blur on attention weights attenuates the curvature of the energy function, we propose Smoothed Energy Guidance (SEG) in this section. For brevity, we redefine the unconditional score prediction as $\mathbf{s}_\theta(\mathbf{x}, t)$, and the unconditional score prediction with the energy curvature reduced as $\tilde{\mathbf{s}}_\theta(\mathbf{x}, t)$. Specifically, $\tilde{\mathbf{s}}_\theta(\mathbf{x}, t)$ is the prediction with the attention weights blurred using a 2D Gaussian filter $G$ with the standard deviation $\sigma$. We formulate the process as:

$$(\mathbf{Q}\mathbf{K}^\top)_{\text{seg}} = G * (\mathbf{Q}\mathbf{K}^\top), \tag{6}$$

where $*$ denotes the 2D convolution operator. Then, we replace the original attention weights with $(\mathbf{Q}\mathbf{K}^\top)_{\text{seg}}$ and compute the final value as in ordinary self-attention.

For practical purposes when the number of tokens is large, we propose an efficient computation of (6) using the property of a linear map, since the convolution operation is linear. Concretely, blurring queries is exactly the same as blurring the entire attention weights, and we propose the following proposition to justify our claim.

**Proposition 3.1.** *Let* $\mathbf{Q}$ *and* $\mathbf{K}$ *be the query and key matrices in self-attention, and let* $G$ *be a 2D Gaussian filter. Blurring the attention weights with* $G$ *is equivalent to blurring the query matrix* $\mathbf{Q}$ *with* $G$ *and then computing the attention weights.*

*Proof.* Since the convolution operation is linear, we can always find a Toeplitz matrix $\mathbf{B}$ such that:

$$G * (\mathbf{Q}\mathbf{K}^\top) = \mathbf{B}(\mathbf{Q}\mathbf{K}^\top), \tag{7}$$

where $*$ denotes the 2D convolution operation. Using the properties of matrix multiplication, we can rewrite (7) as:

$$\mathbf{B}(\mathbf{Q}\mathbf{K}^\top) = (\mathbf{B}\mathbf{Q})\mathbf{K}^\top = (G * \mathbf{Q})\mathbf{K}^\top. \tag{8}$$

$\square$

Finally, SEG is formulated as follows:

$$d\mathbf{x} = [\mathbf{f}(\mathbf{x}, t) - g(t)^2(\gamma_{\text{seg}}\mathbf{s}_\theta(\mathbf{x}, t) - (\gamma_{\text{seg}} - 1)\tilde{\mathbf{s}}_\theta(\mathbf{x}, t))]dt + g(t)d\bar{\mathbf{w}}, \tag{9}$$

where $\gamma_{\text{seg}}$ denotes the guidance scale of SEG.

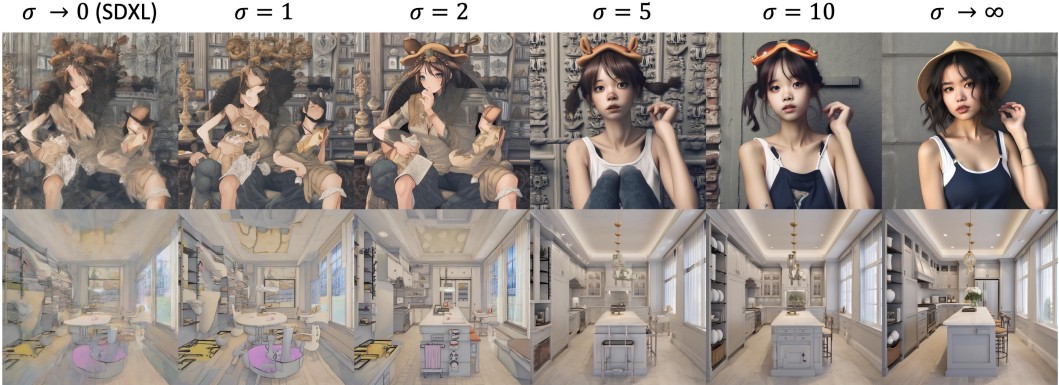

Figure 2: Unconditional generation using SEG.

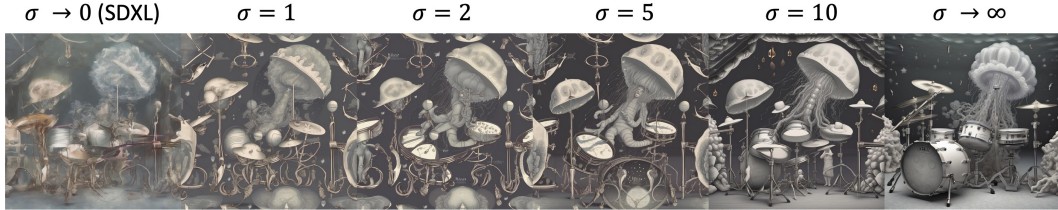

*"a jellyfish playing the drums in an underwater concert"*

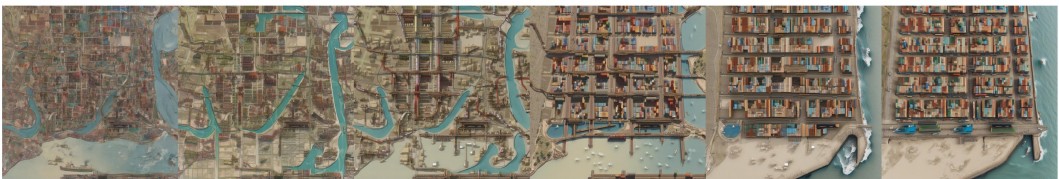

*"a high-resolution satellite image of a bustling shipping port, countless colorful containers"*

Figure 3: Text-conditional generation using SEG.

In a straightforward manner, as SEG does not rely on external conditions, it can be used for conditional sampling strategies such as CFG [14] and ControlNet [51]. For the combinatorial sampling with CFG, following [17], we simply extend (9) for improved conditional sampling with both SEG and CFG as follows:

$$d\mathbf{x} = [\mathbf{f}(\mathbf{x}, t) - g(t)^2((1 - \gamma_{\text{cfg}} + \gamma_{\text{seg}})\mathbf{s}_\theta(\mathbf{x}, t) + \gamma_{\text{cfg}}\mathbf{s}_\theta(\mathbf{x}, t, c) - \gamma_{\text{seg}}\tilde{\mathbf{s}}_\theta(\mathbf{x}, t))]dt + g(t)d\bar{\mathbf{w}}, \quad (10)$$

which is an intuitive result, as the update rule moves $x$ towards the conditional prediction while keeping it far from the prediction with blurred attention weights.

We are likely to get a result with saturation when using a large guidance scale, such as with classifier-free guidance (CFG) [14], self-attention guidance (SAG) [17], and perturbed attention guidance (PAG) [1]. This is a significant caveat since we need to increase the scale to achieve a maximum effect with these methods. Contrary to this, we can fix the scale of SEG as justified in Sec. 5.5 and control its maximum effect through $\sigma$ of the Gaussian blur, making the choice more flexible. For $\sigma$, two extreme cases are recognized. If $\sigma \to 0$, the blurred attention weights remain the same as the original, while when $\sigma \to \infty$, the attention weights merely adopt a single mean value across spatial axes. We find that even the latter extreme case results in a high-quality outcome, corroborating that we can control the quality to the limit without saturation.

## 4 Discussion on related work

Classifier-free guidance (CFG) [14], first proposed as a replacement for classifier guidance (CG) [8] is controlled by a scale parameter. The higher we set classifier-free guidance, the more we get faithful, high-quality images. However, it requires external labels, such as text [30] or class [8] labels, making it impossible to apply to unconditional diffusion models. Also, it requires specific traning procedure with label dropping and it is known that high CFG causes saturation [42].

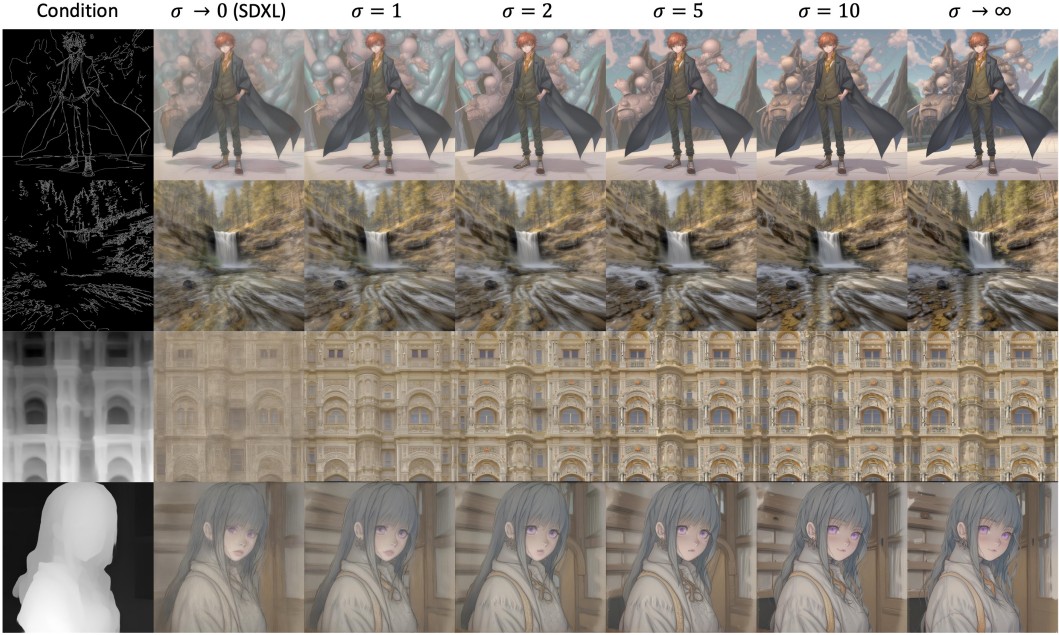

| Condition | $\sigma \to 0$ (SDXL) | $\sigma = 1$ | $\sigma = 2$ | $\sigma = 5$ | $\sigma = 10$ | $\sigma \to \infty$ |

Figure 4: Conditional generation using ControlNet [51] and SEG.

Table 1: Quantitative comparison of SEG with vanilla SDXL [35], SAG [17], and PAG [1] for unconditional generation.

| Metric | Vanilla SDXL [35] | SAG [17] | PAG [1] | SEG $\sigma = 10$ | SEG $\sigma \to \infty$ |
|---|---|---|---|---|---|
| FID↓ | 129.496 | 106.683 | 105.271 | 95.316 | **88.215** |
| LPIPS$_{\text{vgg}}$ ↓ | - | 0.706 | 0.542 | **0.522** | 0.536 |
| LPIPS$_{\text{alex}}$ ↓ | - | 0.644 | 0.472 | **0.454** | 0.472 |

Tackling the caveats of CFG, unconditional approaches such as self-attention guidance (SAG) [17] and perturbed attention guidance (PAG) [1] have been proposed. SAG selectively blurs images with the mask obtained from the attention map and guides the generation process given the prediction. This indirect approach causes saturation and noisy images when given a large guidance scale, leading to the selection of a guidance scale less than or equal to 1. PAG guides images using prediction with identity attention, where the attention map is an identity matrix. However, the reliance on heuristics to make perturbed predictions results in unintended side effects. As an example of the side effects of replacing the attention map with identity attention, PAG changes the visual structure and color distribution of an image, as evidenced in Figs. 5, 8, and 9.

Contrary to these, we control the effect of SEG through the standard deviation of the Gaussian filter, $\sigma$. Moreover, while being theory-inspired, SEG is relatively free from unintended effects. In the following section, we corroborate our claim with extensive experiments.

## 5 Experiments

### 5.1 Implementation details

We build upon the current open-source state-of-the-art diffusion model, Stable Diffusion XL (SDXL) [35], as our baseline, and do not change the configuration. To sample with SEG, we choose the same attention layers (mid-blocks) and guidance scale as PAG [1]. For SEG and PAG sampling, we use the Euler discrete scheduler [21], while for SAG [17], we instead use the DDIM scheduler [45] since the current implementation of SAG does not support the Euler discrete sampler. For SAG and PAG, we use the same configurations they used in the experiments with the previous version of Stable Diffusion, with guidance scales of 1.0 and 3.0, respectively. We set $\gamma_{\text{seg}}$ to 3.0, except in the ablation study.

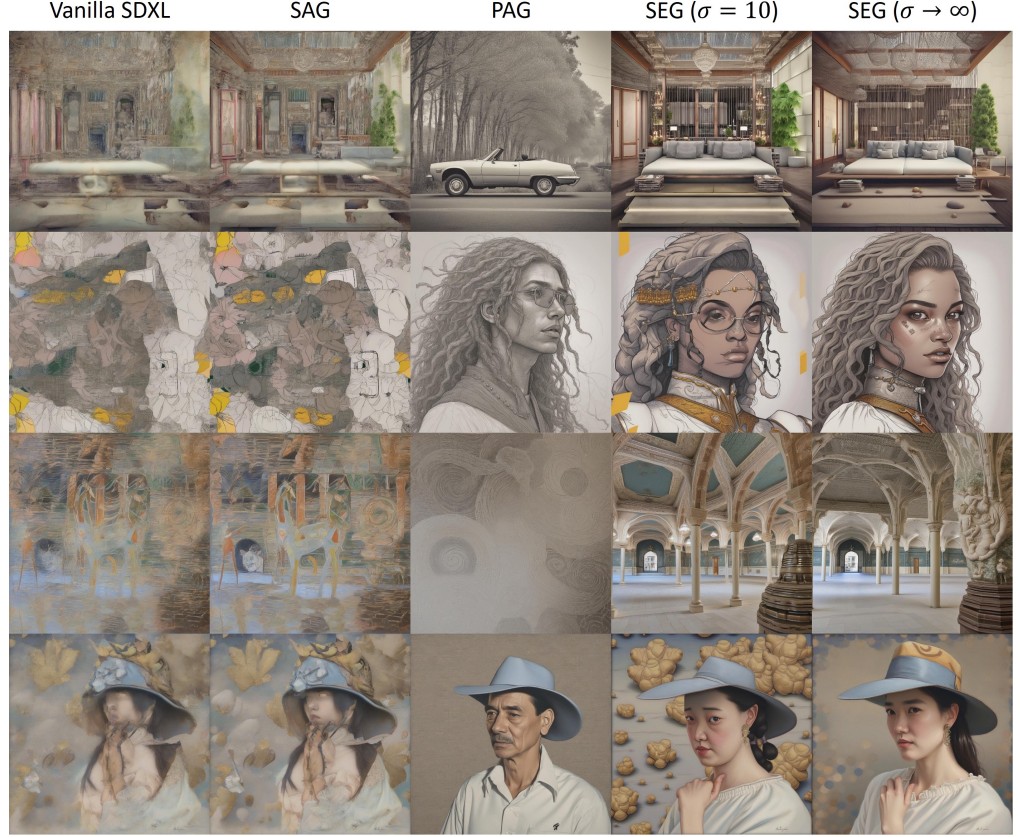

Figure 5: Qualitative comparison of SEG with vanilla SDXL [35], SAG [17], and PAG [1].

Table 2: Text-conditional sampling with different $\sigma$.

| Metric | Vanilla SDXL [35] | SEG | | | | |
|---|---|---|---|---|---|---|
| | | 1 | 2 | 5 | 10 | $\infty$ |
| FID↓ | 53.423 | 48.284 | 41.784 | 33.819 | 29.325 | **26.169** |
| CLIP Score↑ | 0.271 | 0.273 | 0.278 | 0.285 | 0.290 | **0.292** |
| LPIPS$_{vgg}$ ↓ | - | **0.361** | 0.410 | 0.449 | 0.472 | 0.493 |
| LPIPS$_{alex}$ ↓ | - | **0.295** | 0.347 | 0.390 | 0.416 | 0.440 |

## 5.2 Metrics

We use various metrics to evaluate quality (FID [10] and CLIP score [37], calculated with 30k references from the MS-COCO 2014 validation set [28]) and to assess the extent of change due to applied guidance (LPIPS$_{vgg, alex}$ [52]). The latter metric, calculated using the outputs of vanilla SDXL, measures the extent of side effects by comparing guided images to their unguided counterparts.

## 5.3 Controlling image generation with the standard deviation

In this section, our aim is to demonstrate that with SEG, we can sample plausible images using vanilla SDXL [35] under various conditions and even without any conditions, as demonstrated in Fig. 1. Furthermore, without the risk of saturation, we can control the quality and plausibility of the samples. For the results, we use $\sigma \in \{1, 2, 5, 10\}$. Additionally, as mentioned in Sec. 3.3, we present two extreme cases, $\sigma \to 0$ (vanilla SDXL) and $\sigma \to \infty$ (uniform queries).

**Unconditional generation** In this section, our aim is to demonstrate that with SEG, we can sample plausible images from the unconditional mode of the vanilla SDXL, which was originally trained on a large-scale text-to-image dataset. The results are presented in Fig. 1, Fig. 2, and Table 1. The results show a clear tendency to draw higher quality samples by utilizing the differences between the two energy landscapes with different curvatures derived from self-attention mechanisms.

In Fig. 2 and Fig. 13, we show the effectiveness of generating more plausible images, while vanilla SDXL is unable to generate high-quality images without any conditions. The results show a clear tendency to draw higher quality samples by utilizing the differences between the two energy landscapes with different curvatures derived from self-attention mechanisms. When $\sigma$ is larger, the definition and expression of the samples improve, as the difference in curvature becomes more pronounced.

**Conditional generation**   In Figs. 3, 4, 10, 11, and 14, we display sampling results conditioned on text, Canny, and depth map. Using text (Fig. 3), the vanilla SDXL without CFG is unable to generate high-quality images and produces noisy results. Canny and depth map conditioning on SDXL (Fig. 4, 10, and 11) is achieved through ControlNet [51], trained on such maps. The results show that SEG enhances the quality and fidelity of the generated images while preserving the textual and structural information provided by the conditioning inputs. Notably, as $\sigma$ increases, the generated images exhibit improved definition and quality without introducing significant artifacts or deviations from the original condition. The combination with higher CFG scales is shown in Figs. 15–19.

In Table 2, we show the quantitative results for text-conditional generation in terms of $\sigma$. We observe a clear trade-off between image quality (represented by FID and CLIP score) and the deviation from the original sample (represented by LPIPS). We sample 30k images for each $\sigma$ to compute the metrics.

## 5.4   Comparison with previous methods

Since the results are visually favorable when we use $\sigma = 10$ and $\sigma \to \infty$, and they are the best in terms of CLIP score and FID, respectively, we adopt those configurations for comparison of unconditional guidance methods. The results are presented in Figs. 5, 8, 9, and Table 1. Notably, our method achieves better image quality in terms of FID, while remaining similar to the original output of vanilla SDXL as measured by LPIPS, implying a Pareto improvement.

## 5.5   Ablation study

In this section, we address two parameters, $\gamma_{\text{seg}}$ and $\sigma$, and justify that fixing $\gamma_{\text{seg}}$ is a reasonable choice. In Fig. 6, we present the results from our testing. The results reveal that increasing $\gamma_{\text{seg}}$ does not generally lead to improved sample quality in terms of FID and CLIP score, due to various issues such as saturation. In contrast, increasing $\sigma$ tends to improve sample quality and plausibility. This supports the claim that image quality should be controlled by $\sigma$, instead of the guidance scale parameter. We sample 30k images for each combination to calculate the metrics.

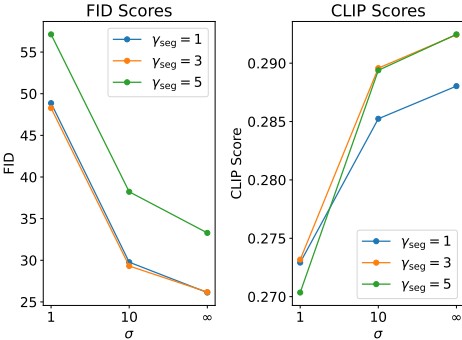

Figure 6: Ablation study on $\gamma_{\text{seg}}$ and $\sigma$.

## 6   Conclusion, limitations and societal impacts

**Conclusion**   We introduce Smoothed Energy Guidance (SEG), a novel training- and condition-free guidance method for image generation with diffusion models. The key advantages of SEG lie in its flexibility and the theoretical foundation, allowing us to significantly enhance sample quality without side effects by adjusting the standard deviation of the Gaussian filter. We hope our method inspires further research on improving generative models, and extending the approach beyond image generation, for example, to video or natural language processing.

**Limitations and societal impacts**   The paper proposes guidance to enhance quality outcomes. Consequently, the attainable quality of our approach is contingent upon the baseline model employed. Furthermore, the application of SEG to temporal attention mechanisms in video or multi-view diffusion models is not addressed, remaining a promising avenue for future research. It is important to note that the improvements achieved through this method may potentially lead to unintended negative societal consequences by inadvertently amplifying existing stereotypes or harmful biases.

## Acknowledgements

I would like to express my gratitude to Yong-Hyun Park, Junha Hyung, and Donghoon Ahn for their valuable feedback and insights. Their thoughtful comments and suggestions have been instrumental in improving this work.

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

# A Full proofs

## A.1 Proof of Lemma 3.1

Let $a_{(i,j)}$ denote the original attention weights and $\tilde{a}_{(i,j)}$ denote the blurred attention weights, as in the main paper. Assume that the original attention weights are properly padded to maintain consistent statistics. Then, the following shows that the mean of the blurred attention weights remains the same.

$$\mathbb{E}_{i,j}[\tilde{a}_{(i,j)}] = \frac{1}{HW} \sum_{i=1}^{H} \sum_{j=1}^{W} \tilde{a}_{(i,j)} = \frac{1}{HW} \sum_{i=1}^{H} \sum_{j=1}^{W} \sum_{m=-k}^{k} \sum_{n=-k}^{k} G(m,n) \cdot a_{(i+m,j+n)}$$

$$= \sum_{m=-k}^{k} \sum_{n=-k}^{k} G(m,n) \cdot \left( \frac{1}{HW} \sum_{i=1}^{H} \sum_{j=1}^{W} a_{(i+m,j+n)} \right)$$

$$= \sum_{m=-k}^{k} \sum_{n=-k}^{k} G(m,n) \cdot \mathbb{E}_{i,j}[a_{(i,j)}] = \mathbb{E}_{i,j}[a_{(i,j)}] \cdot \sum_{m=-k}^{k} \sum_{n=-k}^{k} G(m,n) = \mathbb{E}_{i,j}[a_{(i,j)}]$$

In addition, the variance of the blurred attention weights is smaller than or equal to the variance of the original attention weights.

$$\text{Var}_{i,j}[\tilde{a}_{(i,j)}] = \frac{1}{HW} \sum_{i=1}^{H} \sum_{j=1}^{W} (\tilde{a}_{(i,j)} - \mathbb{E}_{i,j}[\tilde{a}_{(i,j)}])^2$$

$$= \frac{1}{HW} \sum_{i=1}^{H} \sum_{j=1}^{W} \left( \sum_{m=-k}^{k} \sum_{n=-k}^{k} G(m,n) \cdot (a_{(i+m,j+n)} - \mathbb{E}_{i,j}[a_{(i,j)}]) \right)^2$$

$$= \sum_{m=-k}^{k} \sum_{n=-k}^{k} \sum_{r=-k}^{k} \sum_{s=-k}^{k} G(m,n) \cdot G(r,s) \cdot \text{Cov}[a_{(i+m,j+n)}, a_{(i+r,j+s)}]$$

Using the Cauchy-Schwarz inequality and the normalization property of the 2D Gaussian filter, we can show that the variance monotonically decreases when we apply Gaussian blur.

$$\text{Var}_{i,j}[\tilde{a}_{(i,j)}] \leq \sum_{m=-k}^{k} \sum_{n=-k}^{k} \sum_{r=-k}^{k} \sum_{s=-k}^{k} G(m,n) \cdot G(r,s) \cdot \sqrt{\text{Var}[a_{(i+m,j+n)}] \cdot \text{Var}[a_{(i+r,j+s)}]}$$

$$= \left( \sum_{m=-k}^{k} \sum_{n=-k}^{k} G(m,n) \cdot \sqrt{\text{Var}[a_{(i+m,j+n)}]} \right)^2$$

$$\leq \left( \sum_{m=-k}^{k} \sum_{n=-k}^{k} G(m,n) \right) \cdot \left( \sum_{m=-k}^{k} \sum_{n=-k}^{k} G(m,n) \cdot \text{Var}[a_{(i+m,j+n)}] \right)$$

$$= \sum_{m=-k}^{k} \sum_{n=-k}^{k} G(m,n) \cdot \text{Var}[a_{(i+m,j+n)}]$$

$$= \text{Var}_{i,j}[a_{(i,j)}]$$

$\square$

## A.2 Proof of Lemma 3.2

Applying the second-order Taylor series approximation of $e^x$ to our function $f$ around the mean $\mu$, we get:

$$\sum_{i=1}^{H} \sum_{j=1}^{W} e^{a_{(i,j)}} \approx \sum_{i=1}^{H} \sum_{j=1}^{W} \left( e^{\mu} + e^{\mu}(a_{(i,j)} - \mu) + \frac{1}{2} e^{\mu}(a_{(i,j)} - \mu)^2 \right) \tag{11}$$

$$= HW \cdot e^{\mu} + \frac{1}{2} e^{\mu} \sum_{i=1}^{H} \sum_{j=1}^{W} (a_{(i,j)} - \mu)^2 \tag{12}$$

In the last step, we used the fact that $\sum_{i=1}^{H}\sum_{j=1}^{W}(a_{(i,j)} - \mu) = 0$ because $\mu$ is the mean.

Similarly,

$$\sum_{i=1}^{H}\sum_{j=1}^{W} e^{\tilde{a}_{(i,j)}} \approx HW \cdot e^{\mu} + \frac{1}{2}e^{\mu}\sum_{i=1}^{H}\sum_{j=1}^{W}(\tilde{a}_{(i,j)} - \mu)^2 \tag{13}$$

Since $\text{Var}[a] > \text{Var}[\tilde{a}]$, we have:

$$\sum_{i=1}^{H}\sum_{j=1}^{W}(a_{(i,j)} - \mu)^2 \geq \sum_{i=1}^{H}\sum_{j=1}^{W}(\tilde{a}_{(i,j)} - \mu)^2 \tag{14}$$

Therefore, the second-order approximation of $\text{lse}(\mathbf{a})$ is larger than that of $\text{lse}(\tilde{\mathbf{a}})$.

Note that this fact also implies blurring with a Gaussian filter with a bigger variance causes more decrease in the variance of attention weights, because Gaussian filter with a larger variance can always be represented as a convolution of two filters with smaller variances, and the convolution operation is associative.

To find the maximum value subject to the constraint $a_{(1,1)} + a_{(1,2)} + \ldots + a_{(H,W)} = c$ for some constant $c$, we introduce Lagrange multipliers. Let $g(a_{(1,1)}, a_{(1,2)}, \ldots, a_{(H,W)}) = a_{(1,1)} + a_{(1,2)} + \ldots + a_{(H,W)}$. The Lagrangian function is defined as:

$$L(a_{(1,1)}, a_{(1,2)}, \ldots, a_{(H,W)}, \lambda) = e^{a_{(1,1)}} + e^{a_{(1,2)}} + \ldots + e^{a_{(H,W)}} - \lambda(a_{(1,1)} + a_{(1,2)} + \ldots + a_{(H,W)} - c) \tag{15}$$

Taking partial derivatives and setting them to zero yields:

$$\frac{\partial L}{\partial a_{(i,j)}} = e^{a_{(i,j)}} - \lambda = 0 \tag{16}$$

Solving for $a_{(i,j)}$, we obtain $a_{(i,j)} = \ln(\lambda)$ for all $i = 1, 2, \ldots, H$ and $j = 1, 2, \ldots, W$ Summing these equations results in:

$$\lambda = e^{\frac{c}{HW}} \tag{17}$$

Substituting $\lambda$ back into $a_{(i,j)} = \ln(\lambda)$ gives $a_{(1,1)} = a_{(1,2)} = \ldots = a_{(H,W)} = \frac{c}{HW}$. Therefore, the minimum value of $\sum_{i=1}^{H}\sum_{j=1}^{W} e^{a_{(i,j)}}$ is achieved when $a_{(1,1)} = a_{(1,2)} = \ldots = a_{(H,W)}$. $\qquad\square$

### A.3  Proof of Theorem 3.1

Let $\mathbf{a} = (a_1, \ldots, a_n)$ denote the attention values before the softmax operation, and let $\tilde{\mathbf{a}} = (\tilde{a}_1, \ldots, \tilde{a}_n)$ denote the attention values after applying the 2D Gaussian blur. Let $\mathbf{H}$ denote the Hessian of the original energy, $i.e.$, the derivative of the negative softmax, and $\tilde{\mathbf{H}}$ denote the Hessian of the underlying energy associated with the blurred weights.

The elements in the $i$-th row and $j$-th column of the Hessian matrices are given by:

$$h_{ij} = (\xi(\mathbf{a})_i - \delta_{ij})\xi(\mathbf{a})_j, \tag{18}$$

$$\tilde{h}_{ij} = (\xi(\tilde{\mathbf{a}})_i - \delta_{ij})\xi(\tilde{\mathbf{a}})_j b_{ij}, \tag{19}$$

respectively, where $b_{ij}$ are the elements of the Toeplitz matrix corresponding to the Gaussian blur kernel, and $\delta_{ij}$ denotes the Kronecker delta.

Assuming $\xi(\tilde{\mathbf{a}})_i\xi(\tilde{\mathbf{a}})_j \approx 0$ and $\xi(\mathbf{a})_i\xi(\mathbf{a})_j \approx 0$ for all $i$ and $j$, which is a reasonable assumption when the number of token is large and the softmax values get small, the non-diagonal elements of the Hessians approximate to 0 and the diagonal elements dominate. Therefore, the determinants of the Hessian matrices are approximated as the product of the dominant terms:

$$|\det(\mathbf{H})| \approx \prod_{i=1}^{n}\xi(\mathbf{a})_i, \quad |\det(\tilde{\mathbf{H}})| \approx \prod_{i=1}^{n}\xi(\tilde{\mathbf{a}})_i b_{ii} \tag{20}$$

We have the following inequality:

$$\prod_{i=1}^{n} \xi(\tilde{\mathbf{a}})_i b_{ii} < \prod_{i=1}^{n} \xi(\tilde{\mathbf{a}})_i = \frac{e^{\sum_{j=1}^{n} \tilde{a}_j}}{(\sum_{j=1}^{n} e^{\tilde{a}_j})^n} \tag{21}$$

$$\leq \frac{e^{\sum_{j=1}^{n} a_j}}{(\sum_{j=1}^{n} e^{a_j})^n} = \prod_{i=1}^{n} \xi(\mathbf{a})_i, \tag{22}$$

where the first inequality follows from the property of the Gaussian blur kernel, $0 \leq b_{ii} < 1$, and the second inequality is derived from Lemmas 3.1 and 3.2, which demonstrate the mean-preserving property and the decrease in the lse value when applying a blur. The monotonicity of the logarithm function implies that the denominator involving the blurred attention weights is smaller. Eventually, we obtain the following inequality:

$$|\det(\tilde{\mathbf{H}})| < |\det(\mathbf{H})|. \tag{23}$$

This implies that the updated value is derived with attenuated curvature of the energy function underlying the blurred softmax operation compared to that of the original softmax operation. $\qquad\square$

## B  Dual definition

As we previously stated in Section 2.2, we have the dual definition regarding (5), where we use swapped indexing. Importantly, the swapped indices can be interpreted as altering the definition of attention weights to $\mathbf{A} := \mathbf{K}\mathbf{Q}^{\top}$.

A similar conclusion can be drawn as in the main paper, except that query blurring becomes key blurring with this definition. To see this, Eq. 7 changes slightly with this definition, using the symmetry of the Toeplitz matrix $\mathbf{B}$:

$$G * (\mathbf{K}\mathbf{Q}^{\top}) = \mathbf{B}(\mathbf{K}\mathbf{Q}^{\top}) \tag{24}$$

$$= ((\mathbf{K}\mathbf{Q}^{\top})^{\top}\mathbf{B}^{\top})^{\top} \tag{25}$$

$$= (\mathbf{Q}\mathbf{K}^{\top}\mathbf{B}^{\top})^{\top} \tag{26}$$

$$= (\mathbf{Q}(\mathbf{B}\mathbf{K})^{\top})^{\top} \tag{27}$$

$$= (\mathbf{Q}(G * \mathbf{K})^{\top})^{\top} \tag{28}$$

$$= (G * \mathbf{K})\mathbf{Q}^{\top}, \tag{29}$$

where $*$ denotes the 2D convolution operation. Empirically, this altered definition does not introduce a significant difference in the overall image quality, as shown in Fig. 12.

## C  Additional qualitative results

In this section, we present further qualitative results to demonstrate the effectiveness and versatility of our Smoothed Energy Guidance (SEG) method across various generation tasks and in comparison with other approaches.

**Comparison with previous methods**  Figs. 8 and 9 provide a qualitative comparison of SEG against vanilla SDXL [35], Self-Attention Guidance (SAG) [17], and Perturbed Attention Guidance (PAG) [1]. These comparisons highlight the superior performance of SEG in terms of image quality, coherence, and adherence to the given prompts. SEG consistently produces sharper details, more realistic textures, and better overall composition compared to the other methods.

**Conditional generation with ControlNet**  Figs. 10 and 11 showcase the application of SEG in conjunction with ControlNet [51] for conditional image generation. These results illustrate how SEG can enhance the quality and coherence of generated images while maintaining fidelity to the provided control signals. The images demonstrate improved detail, texture, and overall visual appeal compared to standard ControlNet outputs without prompts.

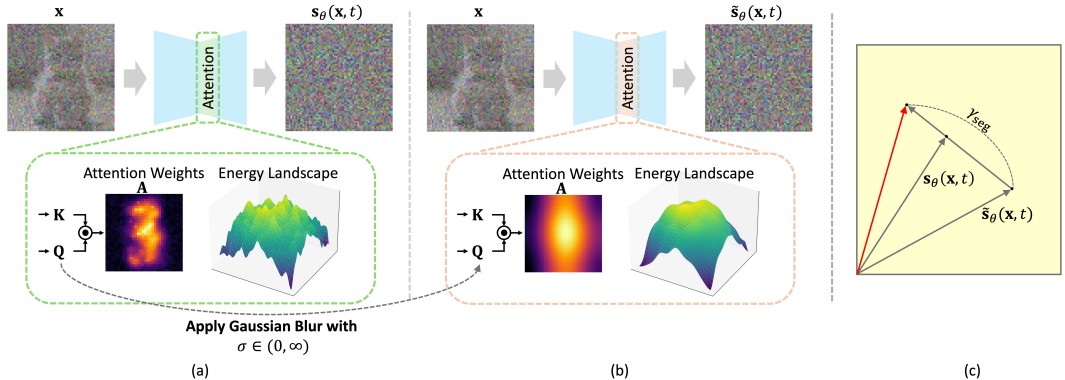

Figure 7: Pipeline of SEG. (a) Original sampling process, self-attention weights, and the corresponding energy landscape. (b) Our modified sampling process with blurred queries where $\sigma \in (0, \infty)$, inducing blurred attention weights and the corresponding smoothed energy landscape. (c) A conceptual figure of $\gamma_{\text{seg}}$. Note that since the guidance linearly extrapolates predictions from (a) and (b), a high guidance scale causes samples to be out of the manifold.

**Unconditional and text-conditional generation**  Fig. 13 demonstrates the capability of SEG in unconditional image generation, showcasing its ability to produce high-quality, diverse images without text prompts. Fig. 14 exhibits text-conditional generation results using SEG, illustrating its effectiveness in translating textual descriptions into visually appealing and accurate images.

**Interaction with classifier-free guidance**  Figs. 15–19 present a series of experiments exploring the combination of SEG with CFG. In these experiments, the SEG guidance scale ($\gamma_{\text{seg}}$) is fixed at 3.0, while the CFG scale is varied. The results demonstrate that SEG consistently improves image quality across different CFG scales without causing saturation or significant changes in the general structure of the images.

**Ablation study**  Fig. 20 displays a visual example of unconditional generation with controlled $\gamma_{\text{seg}}$ and $\sigma$. Consistent with results in Sec. 5.5, controlling image quality with $\sigma$ has fewer side effects than controlling with $\gamma_{\text{seg}}$.

# D  Pipeline figure

The overall pipeline and conceptual framework of SEG are presented in Fig. 7. Fig. 7 (a) and Fig. 7 (b) depict the original sampling process and the modified sampling process with smoothed energy, respectively. Fig. 7 (c) illustrates the the final prediction (the red arrow) with the guidance scale.

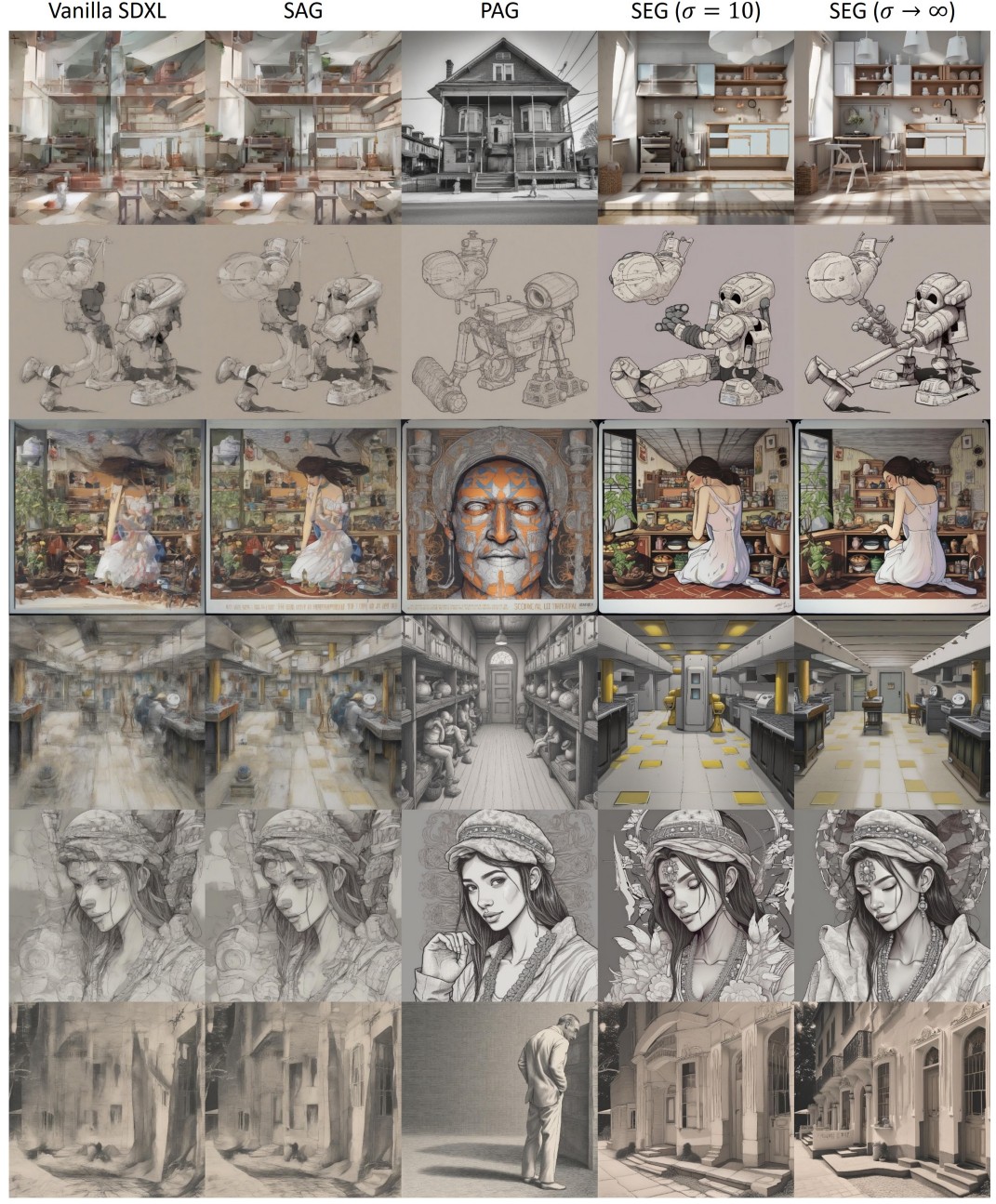

Figure 8: Qualitative comparison of SEG with vanilla SDXL [35], SAG [17], and PAG [1].

Vanilla SDXL   SAG   PAG   SEG ($\sigma = 10$)   SEG ($\sigma \to \infty$)

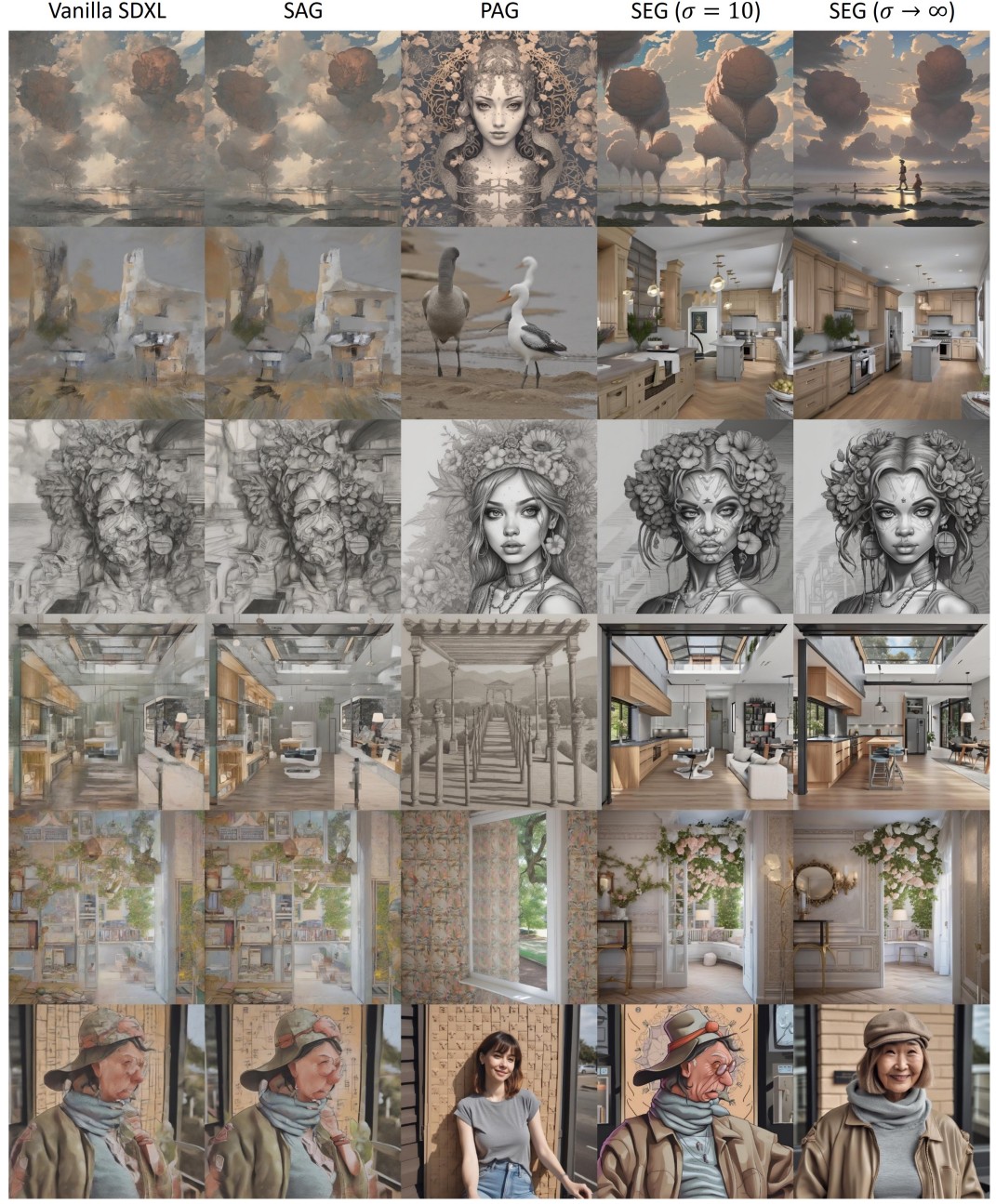

Figure 9: Qualitative comparison of SEG with vanilla SDXL [35], SAG [17], and PAG [1].

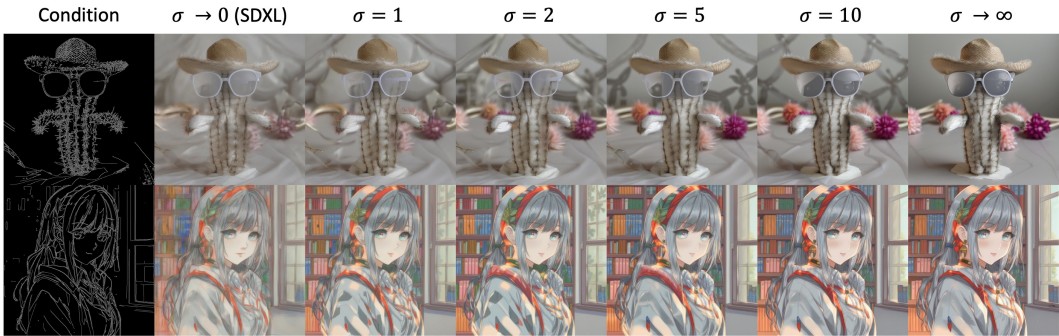

Figure 10: Conditional generation using ControlNet [51] and SEG.

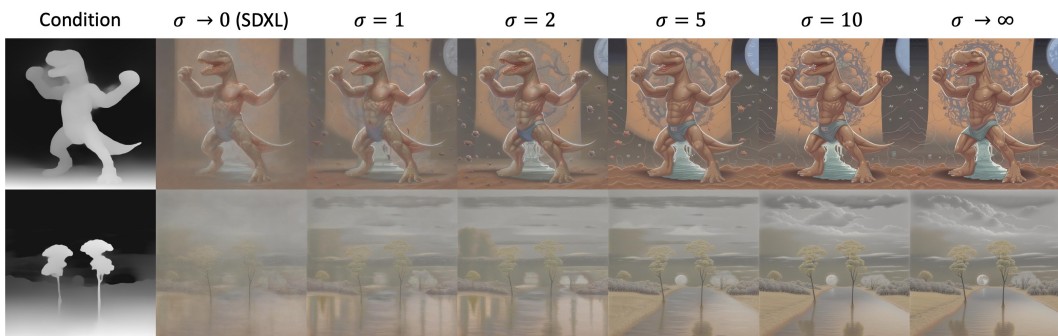

Figure 11: Conditional generation using ControlNet [51] and SEG.

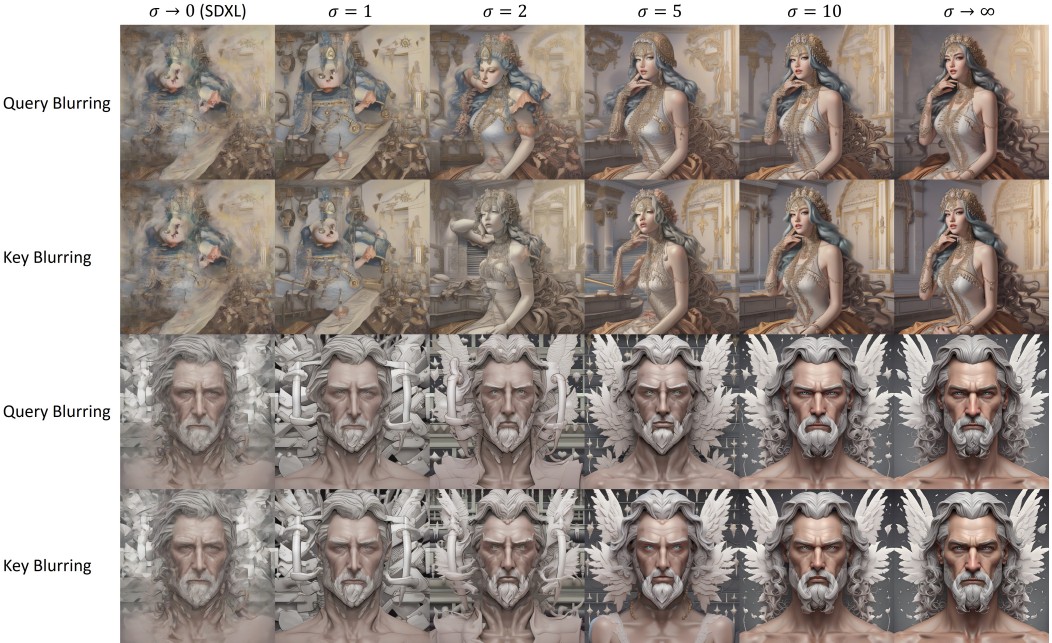

Figure 12: Comparison between query and key blur across different values of $\sigma$.

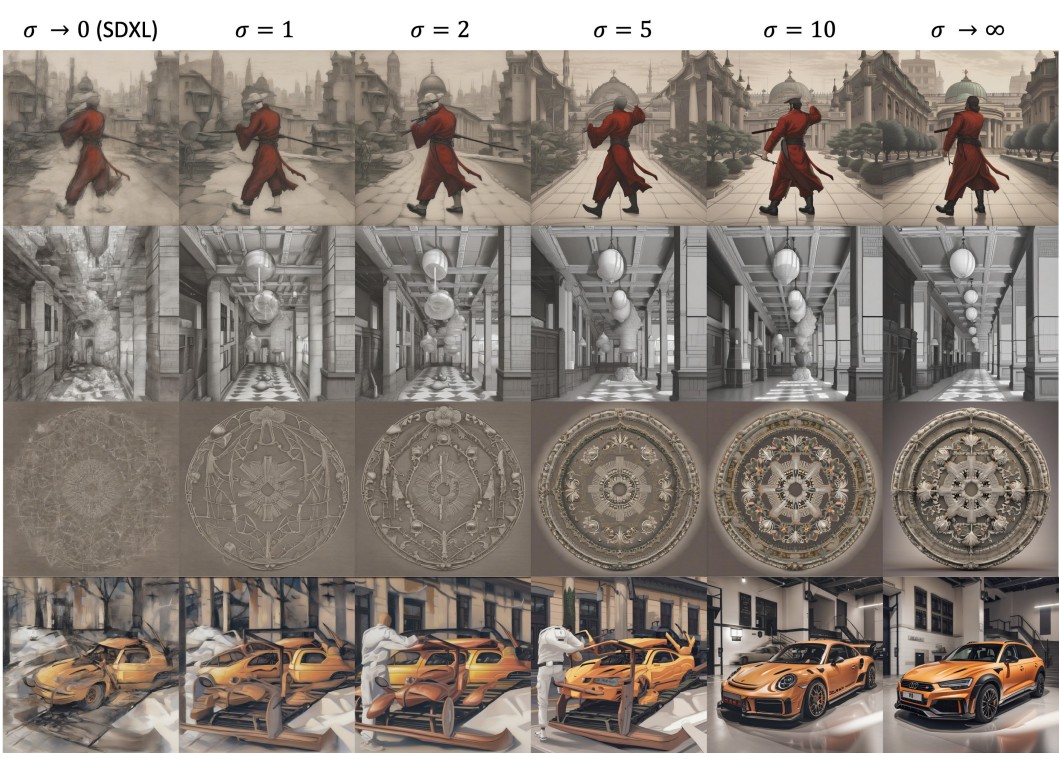

Figure 13: Unconditional generation using SEG.

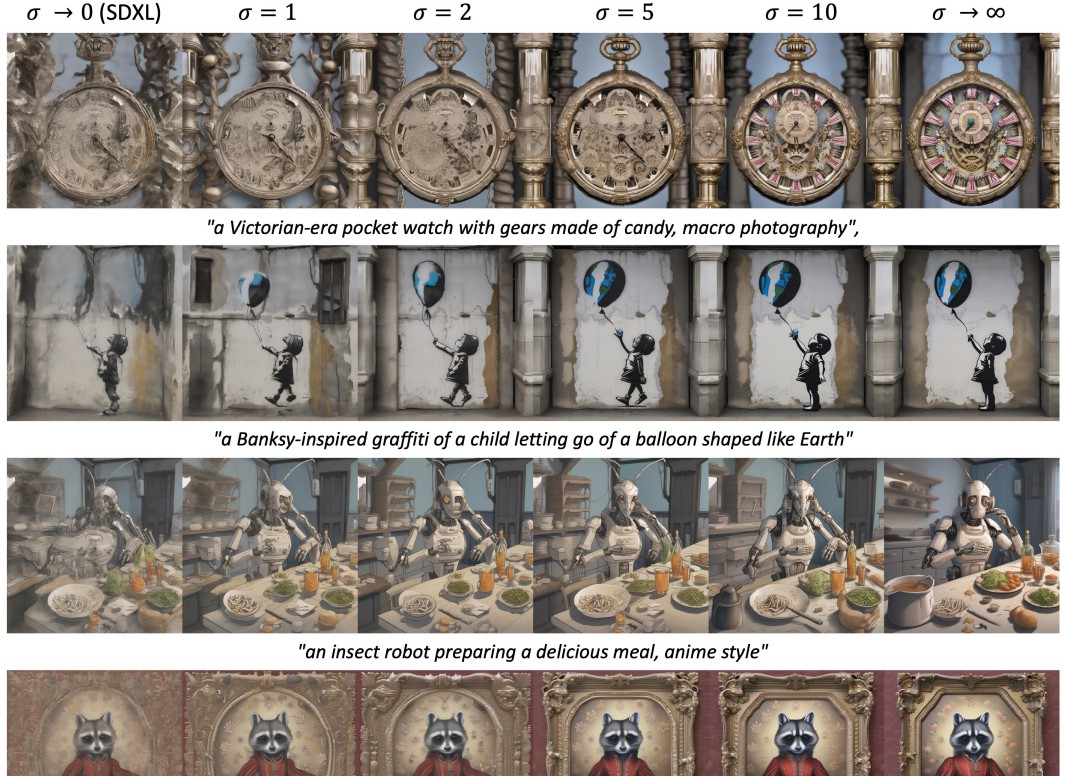

Figure 14: Text-conditional generation using SEG.

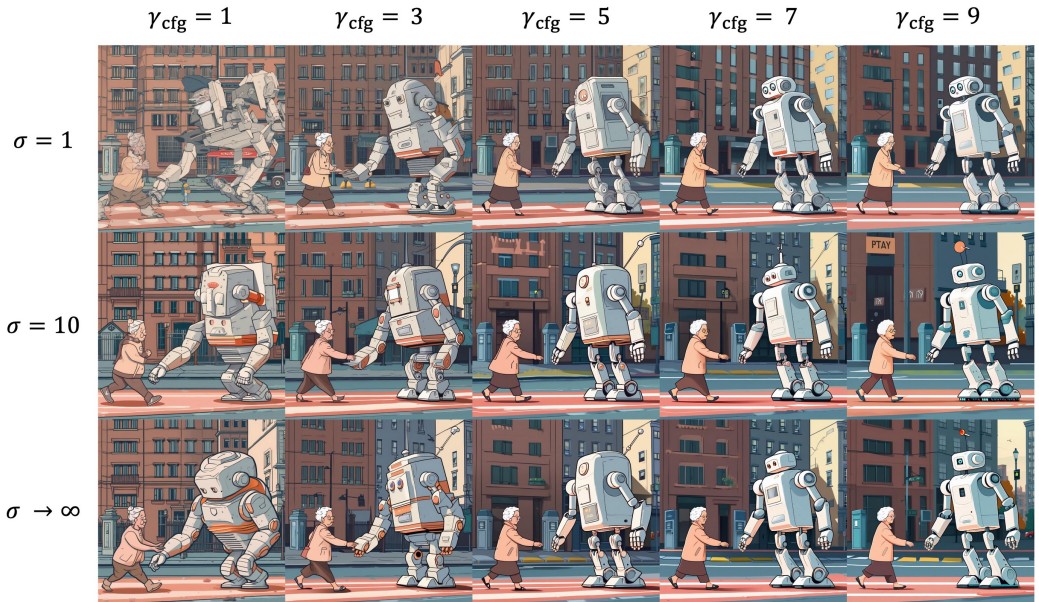

Figure 15: Experiment on the combination of SEG and CFG. $\gamma_{seg}$ is fixed to $3.0$. The prompt is *"a friendly robot helping an old lady cross the street."* Without causing saturation or significant changes in the general structure, SEG improves the image quality.

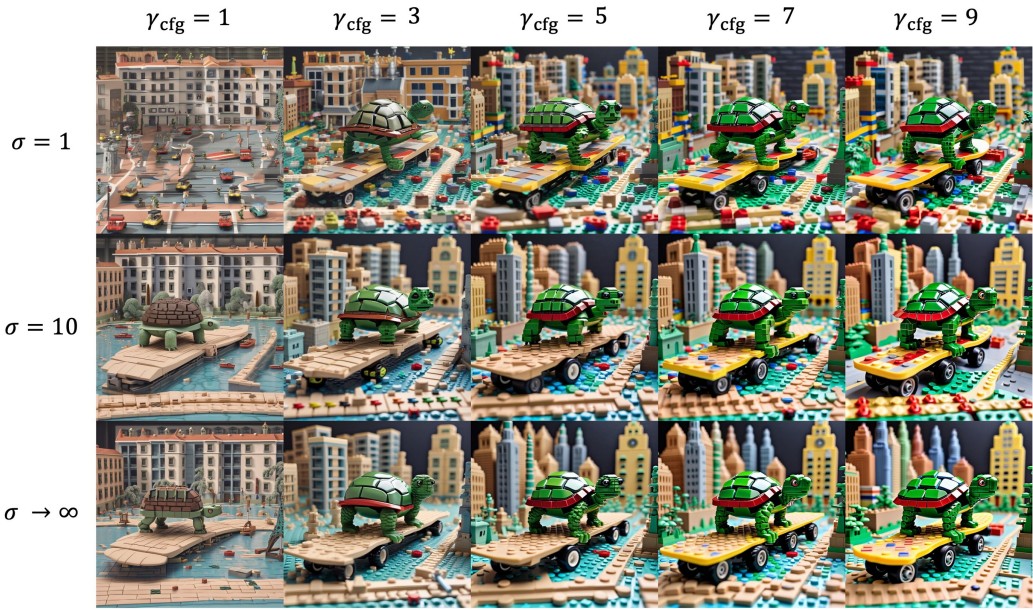

Figure 16: Experiment on the combination of SEG and CFG. $\gamma_{seg}$ is fixed to $3.0$. The prompt is *"a skateboarding turtle zooming through a mini city made of Legos."*

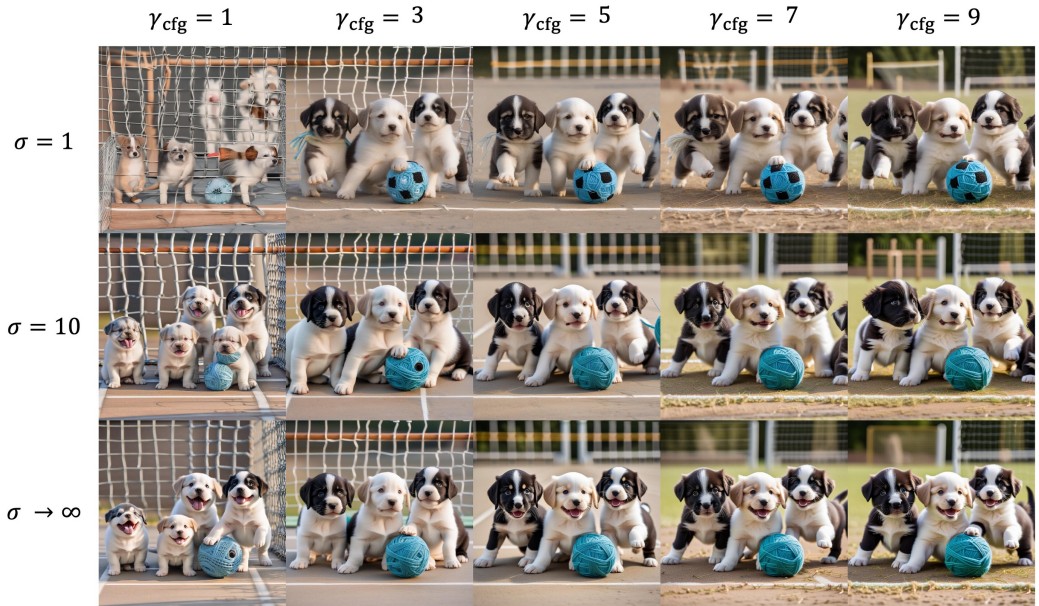

Figure 17: Experiment on the combination of SEG and CFG. $\gamma_{\text{seg}}$ is fixed to 3.0. The prompt is *"a group of puppies playing soccer with a ball of yarn."*

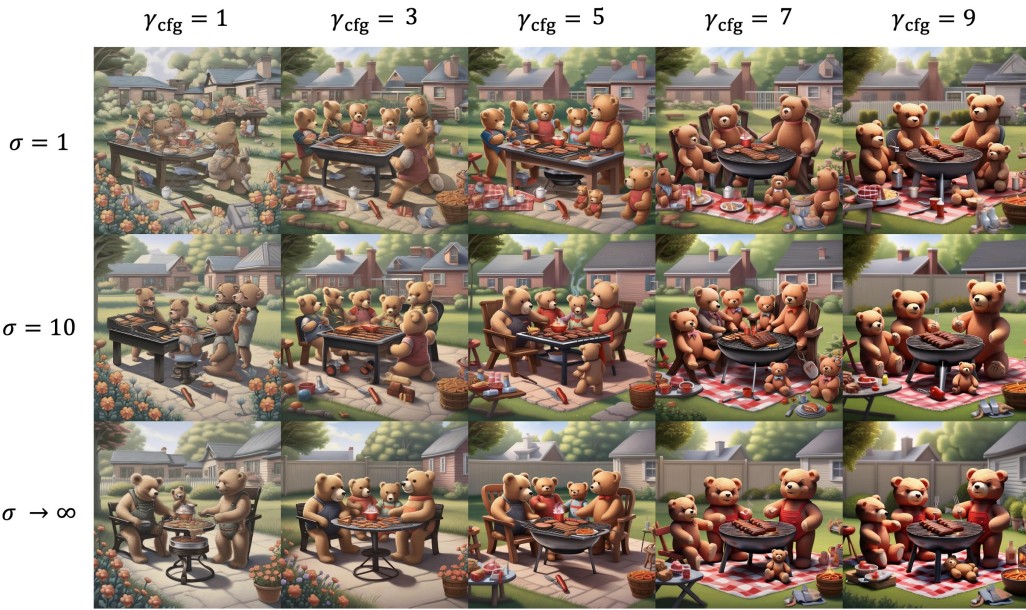

Figure 18: Experiment on the combination of SEG and CFG. $\gamma_{\text{seg}}$ is fixed to 3.0. The prompt is *"a family of teddy bears having a barbecue in their backyard."*

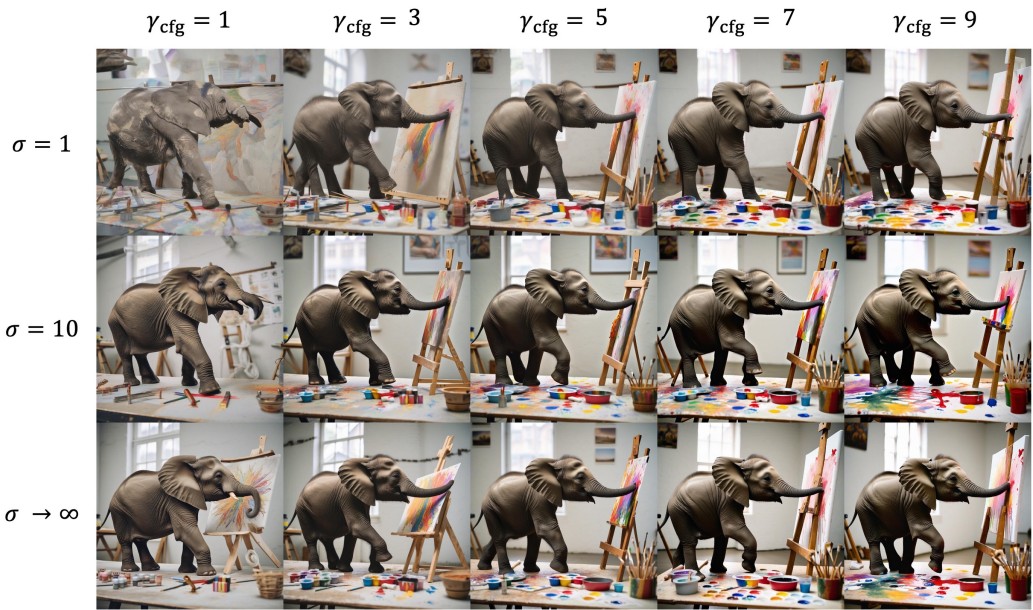

Figure 19: Experiment on the combination of SEG and CFG. $\gamma_{\text{seg}}$ is fixed to 3.0. The prompt is *"a baby elephant learning to paint with its trunk in an art studio."*

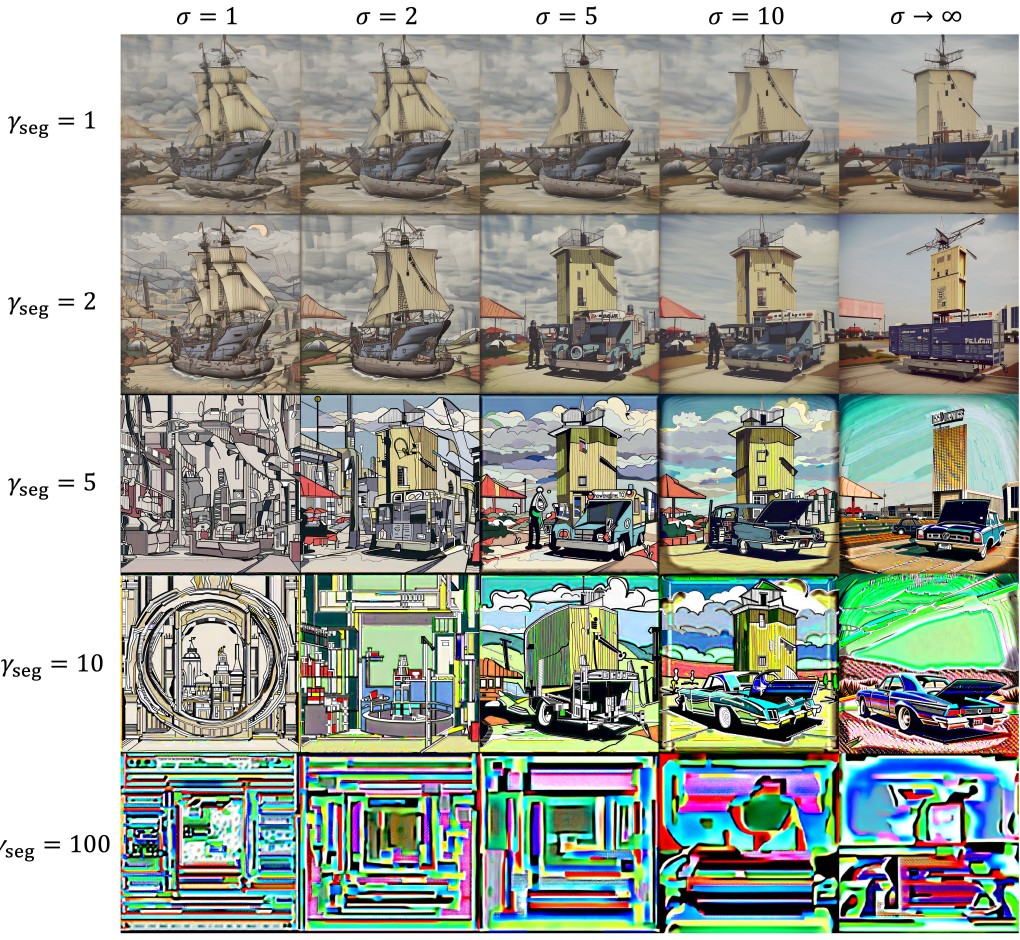

Figure 20: Unconditional generation result with controlled $\gamma_{\text{seg}}$ and $\sigma$.

