# OpenReview forum: "Smoothed Energy Guidance: Guiding Diffusion Models with Reduced Energy Curvature of Attention"
_NeurIPS.cc/2024/Conference — NeurIPS 2024 poster_

### Official Review · Reviewer_cNNp · 2024-07-13

**Soundness:** 2
**Presentation:** 2
**Contribution:** 2
**Rating:** 5
**Confidence:** 1

**Summary:**

The paper presents a condition-free guidance method for diffusion models. The guidance is generated from the self-attention mechanism to perform guidance from an energy-based perspective as an alternative to classifier-free guidance. With this, the work aims to train the models for improved quality performance in conditional and unconditional image generation.

**Strengths:**

The model presents an alternative guidance method for the diffusion model that is independent of an explicit condition, allowing it to work with conditional and unconditional generation.

**Weaknesses:**

As a suggestion by the paper is to refer to a qualitative assessment, what would be the criteria to consider a model or ablation generates a better quality?

**Questions:**

Questions were mentioned in the weaknesses section. Additionally, it would be recommended to introduce the relationship between reducing the curvature of the enemy function and guidance for context.

**Limitations:**

The paper discusses potential limitations and societal impact of generative models in the conclusion section.

---

> ### Author Rebuttal · Authors · 2024-08-05
>
> We would first like to thank the reviewer for acknowledging the strength of our approach in its versatility. We demonstrate SEG's effectiveness in both unconditional and conditional settings, including text-conditional generation and ControlNet conditioning. This flexibility allows SEG to improve image quality across various generation tasks without requiring task-specific modifications. Additionally, we appreciate the reviewer's thoughtful comments and questions. We would like to address the main points raised.
>
> ## Criteria for assessing image quality
> > While we guide readers more towards qualitative assessment, we do employ multiple approaches to evaluate image quality. Quantitatively, we use FID and CLIP scores to measure sample quality. To assess unintended side effects, we utilize LPIPS scores to quantify deviations from unguided images. Qualitatively, we present extensive visual comparisons (e.g., Figs. 2-5 and 7-10 in the main paper) that demonstrate improvements in definition, expression, sharpness of details, realism of textures, and overall composition. We believe this multi-faceted approach provides a comprehensive evaluation of our method's effectiveness. For overall improvement, we have also included uncurated samples from the Vanilla SDXL model, both without and with SEG, in Fig. 2 of the attached PDF.
>
> > Besides, as mentioned in a concurrent work [A], the FD-DINOv2 metric is another means to calculate Fréchet distances and is well-aligned with human perception. For reference, in the table below, we present FD-DINOv2 scores calculated using 50k samples from the EDM2-S model trained on ImageNet-64 to assess fidelity. We also include uncurated qualitative samples from this model in Fig. 3 of the attached PDF. This corroborates how the structure and quality of samples change, as well as the generality of our methods.
> Model | FD-DINOv2$\downarrow$
> --- | ---
> No guidance | 95.1915
> SEG ($\sigma\to\infty$) | **47.4733**
>
> [A] Karras, Tero, et al. "Analyzing and improving the training dynamics of diffusion models." *Proceedings of the IEEE/CVF Conference on Computer Vision and Pattern Recognition*. 2024.
>
> ## Relationship between energy curvature reduction and guidance
> > We appreciate the suggestion to clarify this relationship. On a high level, CFG uses the difference between the prediction based on the sharper conditional distribution and the prediction based on the smoother unconditional distribution to guide the sampling process. By analogy, SEG reduces the curvature of the energy landscape underlying self-attention (Theorem 3.1). This creates a smoother landscape for a minimization step of attention during sampling, analogous to how classifier-free guidance uses the difference between conditional and unconditional distributions. By using the "blunter" prediction from this smoother landscape as negative guidance, SEG enhances sample quality without relying on external conditions or special training.
>
> > From a probabilistic perspective, this process can be thought of as maximizing the likelihood of the attention weights in terms of the Boltzmann distribution conditioned on a given configuration, i.e., the feature map. Blurring the attention weights diminishes this likelihood, as shown in Lemma 3.2, and also reduces the curvature of the distribution, as shown in Theorem 3.1.
>
> ## General response and additional figures
> > We respectfully refer the reviewer to our general response and additional figures provided above. This material addresses key points raised in the initial review and highlights the strengths of our paper, as noted by other reviewers. Additionally, we have included new figures and results that we believe may address your concerns.
>
> We hope these clarifications address the reviewer's concerns and highlight the strengths and contributions of our work. We're happy to provide any additional information or clarifications if needed.

---

> ### Author Response · Authors · 2024-08-12
> **Further Questions Welcome**
>
> Dear Reviewer cNNp,
>
> Thank you again for your time and effort in reviewing our manuscript. We have posted our response addressing your concerns and suggestions.
>
> If you have any additional questions or require further clarification, we are happy to discuss them. We eagerly await your valuable feedback.
>
> Best regards,
>
> Authors of Submission #4721

---

> > ### Comment · Reviewer_cNNp · 2024-08-14
> > **Rensponse to Rebuttal**
> >
> > I thank the authors for the response. I have decided to maintain my score.

---

### Official Review · Reviewer_CTSN · 2024-07-26

**Soundness:** 3
**Presentation:** 3
**Contribution:** 3
**Rating:** 8
**Confidence:** 4

**Summary:**

The paper proposes a technique to improve unconditional sampling from diffusion models. The main idea is to translate the notion of classifier-free guidance (CFG) to the case in which there is no condition available. To this end, the paper notes that the conditional prediction is "sharp", while the unconditional prediction is "smooth", or more simply put, the unconditional prediction is smoother than the conditional one. The paper also notes that applying Gaussian filtering on the attention weights of the model's unconditional prediction yields an (even) smoother prediction. The paper then combines these two observations, and proposes SEG, a version of CFG that is applicable to the unconditional case: CFG requires the conditional and unconditional predictions; SEG replaces the conditional prediction with the unconditional, while replacing the unconditional with its smoothed version. By doing this procedure, SEG manages to preserve the sharpness-smoothness relation that CFG has between the factors (the conditional and unconditional predictions).

The paper then demonstrates that this procedure translates into improved performance compared to reasonable competitors on standard benchmarks.

**Strengths:**

- The paper proposes a simple yet effective procedure to improve unconditional sampling in diffusion models. The idea, in my view, is elegant
- The paper is, over all, well written and clear
- Both qualitative and quantitative evaluations seem to demonstrate the paper's point on the empirical side
- I think the paper's proposal could be used for improving the efficiency of conditional sampling, which is arguably an even more useful case

**Weaknesses:**

## Weaknesses that don't affect my rating
- The Method section could strongly benefit from a figure illustrating on what object, exactly, is the Gaussian blur being applied. The paper currently doesn't have a "pipeline" figure
- I understand the value of Lemma 3.1, but I don't think it's necessary to provide a proof in the main paper. As a side note, I don't even think it's necessary to provide a proof: as far as I understand, this fact is widely known, since the Gaussian filter is normalized and symmetric, no?
  - For instance, I think a well-known intuition in the computer vision community is that (infinite) successive applications of Gaussian blurring on an image result in an image with a single color, that corresponds to the average color of the initial image (i.e. same mean, and zero variance).
- L188 mentions PAG. It would be useful to remind the reader what it stands for. (I think it was originally just mentioned in the introduction)
- Unless I'm misunderstanding, L211 claims that, contrary to two other methods--SAG and PAG--the presented method is both training- and condition-free. I'm not sure I follow: aren't those two methods both also training- and condition-free?
- I understand what the authors mean in L231 by "FID has been found not to be a strict measure of image quality". However, FID in itself does not measure image quality, but distance w.r.t. some distribution
------------
## Weaknesses that don't affect my rating, but should be addressed:
- Eq. (5): please consider explaining that "lse" standards for logsumexp a bit before L85: by then it's a bit confusing already
- Isn't L86 redundant (since the same statement is made--and referenced--in L75?)
- L95: please consider denoting the LSE with some convention: just writing "lse" lends itself to confusion
- L97: a negative prediction of what?
- L108: to make which operation tractable?
- L124: increases the LSE or decreases (according to L127). My intuition is that it decreases, since LSE approximates the maximum of the tensor
- The proof of proposition 3.1 could be made shorter, or sent to the appendix
- L177: "Note that the Gaussian blur can also be applied to K" I presume this is true because of the commutation property of the convolution operation. Perhaps explicitly stating that in the paper would be better

Typos and such:
- L9: actually?
- L39: why "actually"?
- L70: conditioning on
- L72: demonstrates its prevalence? sounds weird
- L111 and equation at L 113: boldface "a"?
- Minor comment for all over the paper: use \eqref for equations
- L230: "how much guided images are altered from unguided ones" sounds a bit weird
- L232: "being more favorable to users" sounds weird
- Fig. 2 and 11: I think there's a repeated sample? (the one indoors)

**Questions:**

- I can understand that the main interest of the paper was improving unconditional generation. The paper achieves this objective by, in a sense, computing a less sharp/more smooth prediction than the unconditional one; this prediction is combined with the original unconditional prediction in the same way that CFG operates. I think that approach is interesting from the point of view of efficiency: a useful prediction is being obtained by an inexpensive operation (i.e. smoothing), instead of another forward pass. Given this context, doesn't it make sense to rather aim at improving conditional generation? (which is arguably the most important one). I think the paper's proposal could be used directly to improve the efficiency conditional generation with CFG: do the conditional forward passes, and replace the unconditional ones with the ones that result from Gaussian smoothing (that is, I think you can replace an entire function evaluation with simple Gaussian smoothing of the conditional prediction). Have the authors considered this option?
[I think Eq. (8) with gamma_seg=0 is somewhat similar, but I'd be thinking of replacing s(x,t) with a \tilde{s}(x,t) = smooth(s(x,t,c))]

**Limitations:**

Yes, they have

---

> ### Author Rebuttal · Authors · 2024-08-05
>
> We sincerely appreciate your acknowledgment of our approach's strengths, particularly its elegant idea and thorough qualitative and quantitative evaluations. Thank you for your careful suggestions. We'd like to address the concerns and questions you've raised:
>
> ## Improving conditional generation
> > We'd like to highlight that our method improves conditional generation even without CFG, thanks to the smoothed energy curvature. When used in its conditional version as a replacement for CFG, rather than being used in combination with CFG, it reduces the number of function evaluations by one. We conducted experiments with the class-conditional EDM2-S model trained on ImageNet-64. For reference, the table below presents FD-DINOv2 scores calculated from 50k samples using this model to assess fidelity. We've also included uncurated qualitative samples in Fig. 3 of the attached PDF.
> | Model | FD-DINOv2$\downarrow$ |
> |-------|-----------|
> | No guidance | 95.1915 |
> | SEG ($\sigma \to \infty$) | **47.4733** |
>
> > It's important to note that our method doesn't propose to blur the output directly, and Gaussian blur on score prediction itself reduces the noise level. Instead, SEG applies Gaussian blur to attention weights. This process still requires a partial forward pass, even when reusing features before the attention weights for efficiency, which incurs slightly more overhead than using the same prediction twice.
>
> > In addition, when it comes to general text-to-image generation like Stable Diffusion, the interface between the given caption and generated image is mostly the cross-attention. However, even though our method can be applied to cross-attention, and even temporal attention (in text-to-video generation, which indeed works to improve consistency), we deal only with self-attention in this paper. We believe the ideas you suggested are promising future directions, though, and those directions will be emphasized in our revised version.
>
> ## Pipeline figure
> > We've included a draft of the pipeline figure (Fig. 4 in the attached PDF) and will incorporate this in the revision. This includes where the Gaussian blur with $\sigma$ is applied, the original and blurred attention weights, the associated energy landscape, and how the score predictions are obtained. We also include in Fig. 4(c) how the linear extrapolation between those predictions in SEG works.
>
> ## Adding intuition to Lemma 3.1
> > While those familiar with Gaussian blur in the computer vision community may intuitively know this fact, we thought a clearer explanation was needed. However, we also find it intuitive that Gaussian blur preserves the mean while decreasing variance, since applying Gaussian blur to images causes the pixels to converge towards a similar value, reducing variance. Therefore, we will consider moving the formal proof to the appendix and instead include this intuition in the main text.
>
> ## Definition of FID
> > We understand your point about FID. While it doesn't directly measure image quality, we use the metric as a proxy for realism and quality because it measures the distance to the real image-text distribution of the COCO dataset. We'll clarify this in the final revision.
>
> ## Other concerns and suggestions
> > We acknowledge the typo in line 211 and confirm that SAG and PAG are indeed training- and condition-free. Also, we've incorporated most of your suggestions to improve the paper's clarity in our revised manuscript.
>
> ## General response and additional figures
> > We respectfully refer the reviewer to our general response and additional figures provided above. This material addresses key points raised in the initial review and highlights the strengths of our paper, as noted by other reviewers. Additionally, we have included new figures and results that we believe may address your concerns.
>
> Thank you again for your valuable feedback. We believe these changes will help us fine-tune the quality and clarity of our paper.

---

> ### Author Response · Authors · 2024-08-12
> **Further Questions Welcome**
>
> Dear Reviewer CTSN,
>
> Thank you again for your time and effort in reviewing our manuscript. We have posted our response addressing your concerns and suggestions.
>
> If you have any additional questions or require further clarification, we are happy to discuss them. We eagerly await your valuable feedback.
>
> Best regards,
>
> Authors of Submission #4721

---

### Official Review · Reviewer_9qC6 · 2024-07-29

**Soundness:** 2
**Presentation:** 3
**Contribution:** 2
**Rating:** 6
**Confidence:** 4

**Summary:**

This paper presents a method for unconditioned image generation based on Diffusion Models, specifically using Stable Diffusion. The proposed method offers an alternative to classifier-free guidance (CFG), eliminating the need to train a classifier for adding conditions. Traditionally, the CFG denoising function incorporates both conditional and unconditional terms, while unconditional generation only involves an unconditional evaluation of the U-Net. Instead, this method performs unconditional diffusion by introducing an additional term to the traditional reverse process.
This new term is an energy-based component calculated as a proportion of the Gaussian blurring of the self-attention layers in the U-Net. The core idea is to use a blurred version of the unconditional self-attention layers to shift the mean distribution of the original unconditional prediction, thereby "smoothing" the prediction.
The paper employs the standard SDXL model and compares its results against the Perturbed Attention Guidance (PAG) and Self-Attention Guidance (SAG) methods.

**Strengths:**

This paper presents an interesting way to use the unconditional prediction of the network as another way to do guidance generation.

This paper presents a method that aims to improve traditional unconditional prediction in diffusion models by smoothing the distribution through the addition of a term to the conventional unconditional image generation process. The proposed idea is analogous to the original CFG methodology in that it combines two different distributions to enhance the generated image. In this case, the smoothing of the distribution is achieved by incorporating a blurred version of the self-attention layers.

**Weaknesses:**

-  Technical correctness of the paper

The claim in lines 191-192, where the paper asserts that $\sigma \rightarrow 0$ remains the same as the original [image], is not entirely clear, and the intuition behind this is not thoroughly presented. The concern arises because Eq. 7 indicates that the proposed method involves adding another term to the original unconditional generation. According to the literature [10], what this paper refers to as $s_\theta(x,t)$ corresponds to a distribution, as does $\tilde{s_{\theta}}(x,t)$. However, having $\sigma$ approaching to 0 does not imply that this term becomes 0; rather, it means that the Gaussian filter affecting the calculation of the self-attention is 0. Therefore, the inference drawn from this reasoning is that if the filter's variance approaches 0, then $s_\theta(x,t)$ approximates to $\tilde{s}_\theta(x,t)$.

Since this paper relies on a strong theoretical background, it is crucial to provide a thorough demonstration and explanation of all claims. For instance, in line 150, what does it mean to be under a "reasonable" assumption to demonstrate the attenuation of the Gaussian curvature? Additionally, what does the paper refer to when discussing the property of linear mapping in the blurring of the queries between lines 167 and 169?

As the method uses a Gaussian kernel for the blurring process, it involves a sigma parameter derived from the Gaussian filter. However, there is insufficient discussion on how this parameter significantly affects the saturation of the generated image. From line 211, it can be interpreted that sigma controls the saturation, but between lines 274-276, the paper indicates that saturation can be altered by using the guidance scale for this method. The question is how sigma and gamma can be balanced to avoid over-saturated images and whether other characteristics of the image (brightness, darkness, vibrance, etc.) can be controlled through these parameters.

- Experimental validation

This paper limits its exploration by not providing further comparisons using other diffusion models as backbones. This omission raises questions about the proposed method's generalization capability.

It is not clear why the paper does not report the perceptual metric for the Vanilla SDXL in Table 1.
The lack of a metric to compare the perceptual performance of the Vanilla SDXL against the proposed method raises a concern, as the validation is then limited to just a few image examples presented in this manuscript. Without this measure, it is difficult to fully understand the overall performance of the proposed method compared to the baseline and other state-of-the-art methods.

The disclaimer in lines 231–232 regarding the FID is acknowledged. However, when it comes to image generation, these are the metrics that the community has adopted and learned to interpret. Of course, this does not limit new research from proposing new ways to measure image generation to provide a "strict measure of image quality" and other attributes.

This paper claims (line 14-15) that its implementation does not significantly increase computational cost. However, it is unclear how the computational requirements should be adjusted to use it. Moreover, the paper directly compares its method against unconditional generation, which only requires a single network evaluation, while this method (according to Eq. 7) requires at least two network evaluations. Furthermore, using a conditional input would necessitate at least three network evaluations, likely leading to a considerable increase in computational cost. It is suggested that the paper include an analysis of the trade-off in terms of the number of function evaluations (NFEs) to fully understand how many network evaluations are needed and how this method impacts computational cost.

It would have been interesting to see how other filter kernels perform besides Gaussian blurring.

- Presentation

Although this paper is well-written and presented, its intelligibility could be improved by adding a few more sentences to clarify some of the intuitions described above.

**Questions:**

The questions are listed in the explanation of the weaknesses. It is encouraged to provide further explanation to the presented concerns to have the chance of increasing this initial score.
In general, the paper, while presenting theoretical justifications, needs clearer explanations of its claims, particularly concerning Gaussian curvature attenuation and the linear mapping property in blurring queries. The impact of the sigma parameter on image saturation requires more discussion, including balancing sigma and gamma. The lack of comparisons with other diffusion models limits the assessment of the method's generalization, and the absence of perceptual metrics for Vanilla SDXL restricts validation to a few examples. While the FID disclaimer is noted, traditional community metrics should be used.

---

> ### Author Rebuttal · Authors · 2024-08-05
>
> We appreciate your thoughtful review of our paper. Below is our response to your review:
>
> ## When sigma approaches 0
> > SEG with $\sigma\to 0$ is equivalent to the original sampling process. This doesn't necessarily mean $\\tilde{s}\_\\theta(x, t)$ goes to 0, but rather that the Gaussian kernel becomes a Dirac delta kernel, implying a single value of 1 in the center after discretization and normalization. Filtering with this kernel is an identity operation, so $s\_\\theta(x, t) = \\tilde{s}\_\\theta(x, t)$ as the filtering has no effect. Replacing $\\tilde{s}\_\\theta(x, t)$ with $s\_\\theta(x, t)$ in Eq. 7 yields:
> $$dx=[f(x,t)-g(t)^2(\gamma_\mathrm{seg}s_\theta (x, t)-(\gamma_\mathrm{seg}-1)s_\theta(x, t))]dt+g(t)d\bar{\omega}=[f(x, t)-g(t)^2(s_\theta (x, t))]dt+g(t)d\bar{\omega},$$
> since the gamma $\gamma_\mathrm{seg} {s}_\theta (x, t)$ terms cancel out each other.
>
> ## Explanation on claims
> > We thank the reviewer for raising this point. Although we already have the basic explanation on claims in the Appendix, we would like to clarify more.
>
> > Let $\mathbf{a}=(a_1,\ldots,a_n)$ denote the attention values before the softmax operation, and let $\tilde{\mathbf{a}} = (\tilde{a}_1,\ldots,\tilde{a}_n)$ denote the attention values after applying the 2D Gaussian blur. Let $H$ denote the Hessian of the original energy, \textit{i.e.}, the derivative of the negative softmax, and $\tilde{{H}}$ denote the Hessian of the underlying energy associated with the blurred weights.
>
> > The elements in the $i$-th row and $j$-th column of the Hessian matrices are given by:
> $$
> h\_{ij}=(\xi(\mathbf{a})_i-\delta\_{ij})\xi(\mathbf{a})_j,\quad\tilde {h}\_{ij}=(\xi(\tilde{\mathbf{a}})_i-\delta\_{ij})\xi(\tilde{\mathbf{a}})_j b\_{ij},
> $$
> respectively, where $b\_{ij}$ are the elements of the Toeplitz matrix corresponding to the Gaussian blur, and $\delta\_{ij}$ denotes the Kronecker delta.
>
> > Assuming $\\xi(\\tilde{a})\_i \\xi(\\tilde{a})\_j \\approx 0$ and $\\xi(a)\_i \\xi(a)\_j \\approx 0$ for all $i$ and $j$, which is a reasonable assumption when the number of tokens is large and the softmax values get small, the non-diagonal elements of the Hessians approximate to 0 and the diagonal elements dominate. The determinants of the Hessian matrices are approximated as:
> $$|\\det(H)|\\approx\\prod\_{i=1}^n\\xi(a)\_i,\\quad   |\\det(\\tilde{H})| \\approx \\prod\_{i=1}^n \\xi(\\tilde{a})\_i b\_{ii}$$
>
> > We have the following inequality:
> $$\\prod\_{i=1}^n\\xi(\\tilde{a})\_i b\_{ii}<\\prod\_{i=1}^n\\xi(\\tilde{a})\_i=(e^{\\sum\_{j=1}^n \\tilde{a}\_j}) / (\\sum\_{j=1}^n e^{\\tilde{a}\_j})^n\\leq (e^{\\sum\_{j=1}^n a\_j})/(\\sum\_{j=1}^ne^{a\_j})^n=\\prod\_{i=1}^n\\xi(a)\_i,$$
> where the first inequality follows from the property of the Gaussian blur kernel, $0 \\leq b\_{ii} < 1$, and the second inequality is derived from Lemmas 3.1 and 3.2, which demonstrate the mean-preserving property and the decrease in the lse value when applying a blur. The monotonicity of the logarithm function implies that the denominator involving the blurred attention weights is smaller. Eventually, we obtain the following inequality:
> $$|\\det(\\tilde{H})|<|\\det(H)|$$
> This implies that the updated value is derived with attenuated Gaussian curvature ($K=\\det(\\tilde{H})$ in our case), of the energy function underlying the blurred softmax operation compared to that of the original softmax operation.
>
> ## Why it does not present perceptual metric for Vanilla SDXL
> > This is because the LPIPS metric is calculated with Vanilla SDXL (line 229), to measure the side effects (how much the sampling process with guidance deviates from those of the original sampling process). You can recognize the perceptual distance as zero, since it does not differ from the original sampling process.
>
> ## Discussion on parameters
> > In Fig. 1 of the attached PDF, we present samples with controlled $\sigma$ and $\gamma_\text{seg}$, which support our experiment in Sec. 5.5 and Fig. 6 and our claim in the main paper. Contributing to saturation, large $\gamma_\text{seg}$ values linearly push the pixel values far from the original prediction.
>
> ## Additional details of quantitative evaluations
> > While we guide readers more towards qualitative assessment, we employ multiple approaches to evaluate image quality. Quantitatively, we use FID and CLIP scores to measure fidelity of samples. To assess unintended side effects, we utilize LPIPS scores to quantify deviations from unguided images. Qualitatively, we present extensive visual comparisons (e.g., Figures 2-5, 7-10) that demonstrate improvements in definition, expression, sharpness of details, realism of textures, and overall composition. This approach demonstrates our paper's main contribution, as Reviewer CTSN has mentioned.
>
> > Besides, as mentioned in a concurrent work [A], the FD-DINOv2 metric is another means to calculate Fréchet distances and is well-aligned with human perception. For reference, in the table below, we present FD-DINOv2 calculated using 50k samples from the EDM2-S model trained on ImageNet-64 to assess fidelity. We also include uncurated qualitative samples from this model in Fig. 3 of the attached PDF.
> | Model | FD-DINOv2$\downarrow$ |
> |-|-|
> | No guidance | 95.1915 |
> | SEG ($\sigma \to \infty$) | **47.4733** |
>
> [A] Karras et al. "Analyzing and improving the training dynamics of diffusion models." CVPR 2024.
>
> ## Computational cost
> > Our NFE is the same as other methods, SAG and PAG, which is 50 per sample. The contribution part in complexity is the blurring operation, which incurs quadratic cost in the number of tokens. The query blurring instead blurs the query matrix rather than the attention map (shown as equivalent in lines 170-176), so it avoids the quadratic complexity and it's our contribution that makes the blurring process feasible in the high resolution situation.
>
> ## General response and additional figures
> > We respectfully refer the reviewer to our general response and additional figures provided above.

---

> ### Author Response · Authors · 2024-08-12
> **Further Questions Welcome**
>
> Dear Reviewer 9qC6,
>
> Thank you again for your time and effort in reviewing our manuscript. We have posted our response addressing your concerns and suggestions.
>
> If you have any additional questions or require further clarification, we are happy to discuss them. We eagerly await your valuable feedback.
>
> Best regards,
>
> Authors of Submission #4721

---

> > ### Comment · Reviewer_9qC6 · 2024-08-12
> > **Final Questions**
> >
> > Dear Authors,
> >
> > Thank you for your response! The initial comments have clarified the concerns regarding Gaussian curvature attenuation, the linear mapping property in blurring queries, and metrics validation. However, I am still curious about whether this methodology generalizes to other diffusion backbones. Additionally, could you please provide some insight into how other filter kernels perform compared to Gaussian blurring?
> >
> > Best,
> > Reviewer 9qC6

---

> ### Author Response · Authors · 2024-08-13
> **Response to the Final Questions**
>
> We appreciate your valuable suggestions and feedback. Here, we provide further explanations to address your questions.
>
> Basically, our main theoretical results do not only apply to SDXL but also to different backbones across various conditions equipped with self-attention mechanisms. In addition, we report experiments using the conditional EDM2-S model [A] trained on ImageNet-64. Fig. 3 in the attached PDF showcases uncurated qualitative samples from this model. We also present FD-DINOv2 scores calculated using 50k samples to assess fidelity to the real image data, which is well-aligned with human perception [A]. These results demonstrate that our method generalizes well to a different backbone, showing significant improvements in sample quality.
> | Method     | FD-DINOv2  |
> |------------|------------|
> | No guidance| 95.1915    |
> | SEG        | **47.4733**|
>
> [A] Karras et al. "Analyzing and improving the training dynamics of diffusion models." CVPR 2024.
>
> Additionally, as per the reviewer's suggestion, we conducted experiments using various filter kernels and calculated the scores using 3k samples with the same random seeds for the EDM2-S model:
> | Filter type | Identity | Bilateral ($\sigma_1 = 1$, $\sigma_2 = 0.5$) | Laplacian | Gaussian ($\sigma = 1$) | Gaussian ($\sigma = 10$) | Gaussian ($\sigma \to \infty$) |
> |-------------|-----------|---------|-------------------|--------------------------------|--------------------------------|---------------------------------------------------------------------|
> | FD-DINOv2   | 230.676   | 226.757                                           | 195.178           | 200.623                        | 190.384                        | **190.089**
>
> $\sigma_1$ and $\sigma_2$ used in the bilateral filter denote the parameters controlling the spatial extent of the filter and the influence of intensity differences, respectively.
>
> In this experiment, SEG using the Gaussian filter outperforms those using different filters. While the other filters enhance the score, SEG is the approach that benefits from our theoretical grounding, allowing us to gradually reduce the energy curvature by increasing a single parameter, $\sigma$, which is the main contribution of our paper.
>
> Thank you again for your thoughtful and valuable questions. We hope this clarification addresses your remaining concerns.

---

### Official Review · Reviewer_Rhhf · 2024-07-29

**Soundness:** 2
**Presentation:** 2
**Contribution:** 2
**Rating:** 5
**Confidence:** 3

**Summary:**

The manuscript introduces SEG, a novel training- and condition-free guidance method for enhancing image generation with diffusion models. The method leverages an energy-based perspective of the self-attention mechanism and introduces a technique to reduce the curvature of the energy landscape of attention, thereby improving the quality of generated images. SEG controls the guidance scale by adjusting a Gaussian kernel parameter, offering a flexible and theoretically grounded approach to unconditional and conditional image generation. The authors validate the effectiveness of SEG through extensive experiments on conditions, showcasing its superiority over existing methods like SAG and PAG in terms of sample quality and reduction of unintended effects. My detailed comments are as follows:

**Strengths:**

This article proposes a novel training- and condition-free image generation method, SEG, which significantly improves diffusion models through the theoretical foundation of smooth energy landscapes and the introduction of Gaussian blur on attention weights and the development of efficient query blur techniques. The quality of the images generated and their superiority verified on multiple conditions.

**Weaknesses:**

The method proposed in this manuscript, SEG, relies heavily on the quality of the baseline model and may amplify biases or harmful stereotypes in existing data.

**Questions:**

(1) The author proposes that the model can generate images without giving any conditions, so how does the model know what to generate?
(2) In Figure 2, as σ increases, the image becomes clearer, but the style is completely different from the initial one. Why is this? How to determine the optimal value of σ?
(3) What are the main application scenarios of this model? How to generate the image that the user wants without giving any conditions?
(4) In Figure 5, Qualitative comparison of SEG with vanilla SDXL, SAG, and PAG. However, the image is labeled as PEG. Please provide a detailed explanation. Also, the full name of PEG is not given.
(5) In the manuscript, Smoothed Energy Guidance (SEG) is defined multiple times. Generally, abbreviations should be defined the first time they appear and used consistently in subsequent content. Please also check other abbreviations.

**Limitations:**

There is a lack of detailed discussion, such as how to determine the value of σ and why increasing the value of γ does not improve the sample quality in terms of FID and CLIP scores.

---

> ### Author Rebuttal · Authors · 2024-08-05
>
> We appreciate your thoughtful review of our paper and the recognition of SEG's strengths. We'd like to address the concerns and questions raised:
>
> ## How the model knows what to generate and why the style is different
> > First, we'd like to note that our method claims an inference time boost like other guidance techniques, e.g., CFG, by modulating and utilizing the modeled distribution. On a high level, samples from the blunter distribution have poor definition, expression, sharpness of details, realism of textures, and overall composition. When we use these as the negative prediction of the score for guidance, we gain samples without such properties.
>
> > Concretely, we can explain how SEG guides the sampling process by drawing an analogy to CFG and from a probabilistic perspective:
>
> > a) Similar to how CFG uses the difference between predictions based on sharper conditional and smoother unconditional distributions, SEG reduces the curvature of the energy landscape underlying self-attention (Theorem 3.1). This creates a smoother landscape for attention minimization during sampling, enhancing sample quality without relying on external conditions or special training.
>
> > b) From a probabilistic perspective, the energy is associated with the likelihood of the attention weights in terms of the Boltzmann distribution conditioned on a given configuration, i.e., the feature map. Blurring the attention weights diminishes this likelihood (Lemma 3.2) and reduces the distribution's curvature (Theorem 3.1).
>
> > The style changes with increasing $\sigma$ you mention seem like changes in the realism of textures or colors, and this is also an effect of using a blunter distribution as negative guidance. Still, our method does not change vanilla SDXL more than PAG or SAG in terms of qualitative and quantitative metrics while achieving a significant quality boost. For example, PAG significantly changes the color, style, and structure of the original SDXL output, as shown in Figs. 5, 7, and 8.
>
> ## Discussion on parameters
> > In Fig. 1 of the attached PDF, we demonstrate experiments with controlled $\sigma$ and $\gamma_\mathrm{seg}$, which support our quantitative experiment and claim in Sec. 5.5 and Fig. 6 in the main paper. Larger $\gamma_\mathrm{seg}$ values can cause side effects such as saturation by pushing pixels linearly, potentially moving them out of the manifold, as can be inferred from Fig. 4(c) of the attached PDF. In contrast, Gaussian blur with $\sigma$ naturally smoothens the energy in diffusion models and yields the score prediction which is more likely to exist on the manifold, causing benign predictions even if $\sigma$ gets infinitely big.
>
> > Also, we chose optimal $\sigma$ values ($10$ and $\infty$) based on perceptual quality (in Figs. 2, 3, and 4) and metrics (FID and CLIP Score in Table 2). We found that in most cases, to generate realistic photos with better structure and composition, $\sigma \to \infty$ is the best choice, while for generating aesthetic images, $\sigma=10$ works well.
>
> ## Application scenarios
> > SEG's main applications include enhancing both conditional and unconditional image generation across various domains. It has many applications in large-scale diffusion models trained on text-image pairs for unconditional and general generation. ControlNet, which uses conditions other than text, is one example. SEG can also be used for inverse problems.
>
> ## Reliance on baseline models
> > While SEG does rely on the baseline model, this is inherent to all guidance methods, which aim to enhance or control the generation process during inference. We have clarified this limitation in the paper. Still, it can be seen as a strength that our theoretical results are general to all models and that SEG is used in both unconditional image generation and conditional generation with various types of input, as acknowledged by Reviewers 2oF8 and cNNp.
>
> ## Abbreviations
> > We apologize for the labeling error in Fig. 5. "PEG" should indeed be "SEG". We will correct this in the final revision and ensure consistent use of abbreviations throughout.
>
> ## General response and additional figures
> > We respectfully refer the reviewer to our general response and additional figures provided above. This material addresses key points raised in the initial review and highlights the strengths of our paper, as noted by other reviewers. Additionally, we have included new figures and results that we believe may address your concerns.

---

> ### Author Response · Authors · 2024-08-12
> **Further Questions Welcome**
>
> Dear Reviewer Rhhf,
>
> Thank you again for your time and effort in reviewing our manuscript. We have posted our response addressing your concerns and suggestions.
>
> If you have any additional questions or require further clarification, we are happy to discuss them. We eagerly await your valuable feedback.
>
> Best regards,
>
> Authors of Submission #4721

---

### Official Review · Reviewer_2oF8 · 2024-07-29

**Soundness:** 2
**Presentation:** 2
**Contribution:** 2
**Rating:** 4
**Confidence:** 4

**Summary:**

The method (SEG) discussed in the paper mainly applies an energy-based optimization on the emerging values in the self-attention to reduce the curvature of the energy landscape of attention, leading to improved image quality and less structural change from the original prediction compared to previous approaches. SEG is training- and condition-free and can be used for both unconditional and conditional sampling strategies. The authors validate the effectiveness of SEG by evaluating generated images with and without text conditions, as well as with ControlNet.

**Strengths:**

The paper looks at the refinement of image generation through emerging values in attention layers, which is very interesting. The paper investigated multiple conditions for the diffusion model they used, and the visualizations are insightful.

**Weaknesses:**

The quantitative evaluations needed to be presented in a more detailed manner.
The number of works they compared against could be much more.
e.g. imagic (Kawar, B., Zada, S., Lang, O., Tov, O., Chang, H., Dekel, T., Mosseri, I. and Irani, M., 2023. Imagic: Text-based real image editing with diffusion models. In Proceedings of the IEEE/CVF Conference on Computer Vision and Pattern Recognition (pp. 6007-6017).), LEdit++ (Brack, M., Friedrich, F., Kornmeier, K., Tsaban, L., Schramowski, P., Kersting, K. and Passos, A., 2024. Ledits++: Limitless image editing using text-to-image models. In Proceedings of the IEEE/CVF Conference on Computer Vision and Pattern Recognition (pp. 8861-8870). ) or collaborative diffusion (Huang, Z., Chan, K.C., Jiang, Y. and Liu, Z., 2023. Collaborative diffusion for multi-modal face generation and editing. In Proceedings of the IEEE/CVF Conference on Computer Vision and Pattern Recognition (pp. 6080-6090).),

**Questions:**

What is the advantage of your method compared to other recent works in text-to-image generation that is mentioned in the weaknesses?

**Limitations:**

The validation can be improved.

---

> ### Author Rebuttal · Authors · 2024-08-05
>
> We would first like to thank the reviewer for acknowledging our visualization with various conditions as insightful and our paper as very interesting. We'd like to address the concerns and questions raised:
>
> ## Additional details of quantitative evaluations
>
> > While we guide readers more towards qualitative assessment, we do employ multiple approaches to evaluate image quality. Quantitatively, we use FID and CLIP scores to measure fidelity of samples. To assess unintended side effects, we utilize LPIPS scores to quantify deviations from unguided images. Qualitatively, we present extensive visual comparisons that demonstrate improvements in definition, expression, sharpness of details, realism of textures, and overall composition. As Reviewer CTSN has mentioned, we believe this approach provides a comprehensive evaluation of our method's effectiveness and demonstrate our paper's point on the empirical side.
>
> > Besides, as mentioned in a concurrent work [A], the FD-DINOv2 metric is another means to calculate Fréchet distances and is well-aligned with human perception. For reference, in the table below, we present FD-DINOv2 scores calculated using 50k samples from the EDM2-S model trained on ImageNet-64 to assess fidelity. We also include uncurated qualitative samples from this model in Fig. 3 of the attached PDF. This corroborates how the structure and quality of samples change, as well as the generality of our methods.
> | Model | FD-DINOv2$\downarrow$ |
> | --- | --- |
> | No guidance | 95.1915 |
> | SEG | **47.4733**
>
> [A] Karras, Tero, et al. "Analyzing and improving the training dynamics of diffusion models." *Proceedings of the IEEE/CVF Conference on Computer Vision and Pattern Recognition*. 2024.
>
> ## Compare against more recent works (e.g. Imagic, LEdits++, collaborative diffusion)
>
> > We respectfully would like to emphasize that our method differs from recent text-based editing works like Imagic, LEdits++, and collaborative diffusion in its generality and goal. Our approach is more general, aiming to improve quality for diffusion models equipped with self-attention, regardless of the presence of text conditions.
>
> > However, we recognize that these text-based editing methods are orthogonal to our method and could be potential applications of our approach. In the revised manuscript, we may include a discussion of the potential for synergistic applications in the future direction part.
>
> ## Address the question about advantages over recent works
>
> > We appreciate the opportunity to clarify this point. While our method is complementary to those text-based editing works, our method features several characteristics:
>
> > 1. Generality: Unlike text-based editing methods, our approach can improve image quality without requiring text prompts, making it applicable to a broader range of scenarios. SEG demonstrates effectiveness in both conditional and unconditional image generation scenarios, making it a flexible solution (Reviewers 2oF8 and cNNp).
> 2. Inference-time quality enhancement: SEG improves the overall definition, expression, sharpness of details, realism of textures, and overall composition of generated images across various conditions, e.g., no condition, text, Canny, or depth map (Reviewers 2oF8 and Rhhf). Text-based editing methods the reviewer mentioned do not target and induce quality improvement.
> 3. Novel approach: Our method, Smoothed Energy Guidance (SEG), presents an innovative, training- and condition-free approach to image generation using diffusion models. It offers an interesting alternative to classifier-free guidance (CFG) by leveraging the self-attention mechanism (Reviewers 2oF8, Rhhf, 9qC6, and CTSN).
> 4. Theoretical foundation: The paper provides a solid theoretical grounding, using an energy-based perspective and the concept of smooth energy landscapes to improve image generation (Reviewers Rhhf and CTSN).
>
> ## General response and additional figures
> > We respectfully refer the reviewer to our general response and additional figures provided above. This material addresses key points raised in the initial review and highlights the strengths of our paper, as noted by other reviewers. Additionally, we have included new figures and results that we believe may address your concerns.
>
> We will elaborate on these points in the revised manuscript to clearly articulate the unique benefits of our approach.

---

> > ### Comment · Reviewer_2oF8 · 2024-08-14
> >
> > I thank the authors who have provided additional quantitative evaluation using FD-DINOv2 scores; the comparison is still limited. It may be worth suggesting they expand their quantitative comparisons to include more baseline methods and metrics.
> > The highlights the authors provided regarding the generality of their approach are very interesting; it would be beneficial to see more direct comparisons with state-of-the-art methods in both conditional and unconditional settings to better contextualize their contributions. Methods like LEdit++ and ControlNet are powerful and intuitive approaches, and if the proposed method holds the generalizability advantage, it can be applied along the SOTA methods and improved.

---

> ### Author Response · Authors · 2024-08-12
> **Further Questions Welcome**
>
> Dear Reviewer 2oF8,
>
> Thank you again for your time and effort in reviewing our manuscript. We have posted our response addressing your concerns and suggestions.
>
> If you have any additional questions or require further clarification, we are happy to discuss them. We eagerly await your valuable feedback.
>
> Best regards,
>
> Authors of Submission #4721

---

> ### Author Response · Authors · 2024-08-14
>
> Thank you for your valuable feedback. We appreciate your suggestions, but we believe there are some misunderstandings that we'd like to address.
>
> 1. Regarding comparisons with state-of-the-art methods, **we have already included experiments using ControlNet in our main paper (Figures 4, 9, and 10)**. These comparisons directly address your concern about contextualizing our contributions against powerful approaches.
>
> 2. In addition, we deliberately did not compare our method with **text-based image editing** approaches like LEdit++ because they do not align with our research goals. Our focus is on different editing paradigms, making such comparisons inappropriate as baselines for our work. This point is also addressed in the rebuttal.
>
> As we are at the end of the reviewer-author discussion period, we respectfully ask that you consider the information already provided in our paper and previous responses. We have addressed many of these points earlier and believe our work stands on its own merits within its intended scope. We hope this clarifies our position and demonstrates that we have indeed addressed many of your concerns within the constraints of our research focus. We urge you to reconsider your evaluation in light of this information.
>
> Again, thank you very much for your thoughtful and valuable questions.
>
> Best regards,
>
> Authors of Submission #4721

---

> > ### Comment · Reviewer_2oF8 · 2024-08-14
> >
> > Thank you for the author's response. I will consider them in my final rating.

---

### Author Rebuttal · Authors · 2024-08-05

## General response to reviewers

> We sincerely thank all the reviewers for their thorough evaluation and insightful feedback on our submission. We appreciate the recognition of our work's strengths and the constructive suggestions to fine-tune our manuscript. Based on the reviews, we'd like to highlight the following strengths of our paper:

> 1. **Novelty**: Our method, Smoothed Energy Guidance (SEG), presents an elegant, training- and condition-free approach to image generation using diffusion models. It offers an interesting alternative to other guidance methods, such as CFG, SAG, and PAG, by leveraging smoothed energy landscape of the self-attention mechanism (**Reviewers 2oF8, Rhhf, 9qC6, and CTSN**).
2. **Theoretical grounding**: The paper provides theoretical grounding, using an energy-based perspective and the concept of smooth energy landscapes to improve image generation (**Reviewers Rhhf and CTSN**).
3. **Versatility**: SEG demonstrates effectiveness in both conditional and unconditional image generation scenarios, making it a flexible solution (**Reviewers 2oF8 and cNNp**).
4. **Empirical validation**: Our work provides both qualitative and quantitative evaluations that demonstrate the method's effectiveness (**Reviewer CTSN**). The reviewers also noted the quality of our generated images, verified across various conditions (**Reviewers 2oF8 and Rhhf**).
5. **Clear presentation**: The paper is generally well-written and clear, with insightful visualizations that help convey our method's effectiveness (**Reviewers 9qC6 and CTSN**).

## Additional figures

> In the attached PDF file, we present qualitative results with controlled $\sigma$ and $\gamma_\mathrm{seg}$ (Fig. 1), which support our quantitative experiment and claim in Sec. 5.5 and Fig. 6 in the main paper. We also present more uncurated results (Fig. 2 and Fig. 3), results from the class-conditional model of EDM2-S (Fig. 3), and a conceptual figure of the overall pipeline (Fig. 4).

---

### Decision · Program_Chairs · 2024-09-25

**Decision:**

Accept (poster)

**Comment:**

This paper received split recommendations after the post-rebuttal discussion. The rebuttal adequately addressed most reviewer concerns about technical correctness and experimental validation. The author-reviewer discussion was engaged, and most remaining doubts were clarified. Overall, the paper presents an interesting technique to improve unconditional sampling from diffusion models that can have a practical impact. Reviewer comments and corrections, particularly those expressed by Reviewer 2oF8, should be carefully included in the final version.